# Observation of absorbing aerosols above clouds over the South-East Atlantic Ocean from the geostationary satellite SEVIRI Part 1: Method description and sensitivity

*Fanny Peers[1], Peter Francis[2], Cathryn Fox[2], Steven J. Abel[2], Kate Szpek[2], Michael I. Cotterell[1,2], Nicholas W. Davies[1,2], Justin M. Langridge[2], Kerry G. Meyer[3], Steven E. Platnick[3], Jim M. Haywood[1,2]*

(1) College of Engineering, Mathematics, and Physical Sciences, University of Exeter, Exeter, UK
(2) Met Office, Exeter, UK
(3) NASA Goddard Space Flight Center, Greenbelt, Maryland, USA

## **Abstract**

High temporal resolution observations from satellites have a great potential for studying the impact of biomass burning aerosols and clouds over the South East Atlantic Ocean (SEAO). This paper presents a method developed to retrieve simultaneously aerosol and cloud properties in aerosol above cloud conditions from the geostationary instrument Meteosat Second Generation/Spinning Enhanced Visible and Infrared Imager (MSG/SEVIRI). The above-cloud Aerosol Optical Thickness (AOT), the Cloud Optical Thickness (COT) and the Cloud droplet Effective Radius (CER) are derived from the spectral contrast and the magnitude of the signal measured in three channels in the visible to shortwave infrared region. The impact of the absorption from atmospheric gases on the satellite signal is corrected by applying transmittances calculated using the water vapour profiles from a Met Office forecast model. The sensitivity analysis shows that a 10% error on the humidity profile leads to an 18.5% bias on the above-cloud AOT, which highlights the importance of an accurate atmospheric correction scheme. *In situ* measurements from the CLARIFY-2017 airborne field campaign are used to constrain the aerosol size distribution and refractive index that is assumed for the aforementioned retrieval algorithm. The sensitivities in the retrieved AOT, COT and CER to the aerosol model assumptions are assessed. Between 09:00-15:00 UTC, an uncertainty of 40% is estimated on the above-cloud AOT, which is dominated by the sensitivity of the retrieval to the single scattering albedo. The absorption AOT is less sensitive to the aerosol assumptions with an uncertainty generally lower than 17% between 09:00-15:00 UTC. Outside of that time range, as the scattering angle decreases, the sensitivity of the AOT and the absorption AOT to the aerosol model increases. The retrieved cloud properties are only weakly sensitive to the aerosol model assumptions throughout the day, with biases lower than 6% on the COT and 3% on the CER. The stability of the retrieval over time is analysed. For observations outside of the backscattering glory region, the time-series of the aerosol and cloud properties are physically consistent, which confirms the ability of the retrieval to monitor the temporal evolution of aerosol above cloud events over the SEAO.

## 1. Introduction

The South East Atlantic Ocean (SEAO) provides a natural laboratory for analysing the full range of aerosol-cloud-radiation interactions. During the fire season, large amounts of particles from African biomass burning are transported above the semi-permanent deck of stratocumulus covering this oceanic region. As a result, an important contrast is expected in the Direct Radiative Effect (DRE) of aerosols (i.e. the direct impact of aerosol scattering and absorption of radiation). On one hand, the aerosol scattering above the ocean typically increases the local albedo which leads to a negative DRE at the top of the atmosphere. On the other hand, the sign of the DRE above clouds depends on the underlying cloud albedo and the aerosol absorption. Positive instantaneous DRE of up to +130W m$^{-2}$ has been observed by satellite instruments over the SEAO (De Graaf et al., 2012; Peers et al., 2015). There are many poorly constrained variables, such as the aerosol and cloud properties, vertical structure of aerosol and clouds (Peers et al., 2016), which result in a large spread in the DRE derived from climate models in this region (Zuidema et al., 2016). In addition, the absorption of radiation by aerosols leads to a modification of the atmospheric stability and consequently on the formation, development and dissipation of clouds, i.e. semi-direct effect. Studies have shown that the overlying African biomass burning aerosols are associated with a cloud thickening (Wilcox, 2010 & 2012). This negative semi-direct effect partly compensates the positive DRE of aerosols above clouds over the SEAO. However, as an aerosol plume moves away from the coast and descends into the boundary layer, the heat due to the aerosol absorption could lead to a reduction of the cloud thickness (Koren et al., 2004). Biomass burning particles may also have indirect effects through their interactions with cloud droplets, leading to a modification of the microphysics of the cloud, its lifetime and precipitations (Twomey, 1974; Rosenfeld, 2000). Recent model studies (Gordon et al., 2018; Lu et al., 2018) suggest that the semi-direct and indirect effects of aerosols dominate the DRE over the SEAO, leading to a regional cooling.

Until recently, there has been a relative dearth of observations of biomass burning above clouds as passive sensor retrievals of aerosol and cloud are generally mutually exclusive. In past studies, biases in cloud properties derived from passive shortwave measurements were expected because the impact of aerosol absorption above clouds was not taken into account in the retrievals (Haywood et al., 2004). Over the last decade, techniques have been developed for the observation of aerosols above clouds. POLDER (Polarization measurements from POLarization and Directionality of the Earth's Reflectances) has been used to detect aerosols above clouds and to characterize the aerosol and the cloud layers by exploiting the sensitivity in polarized measurements (Waquet et al., 2013a & 2013b; Peers et al., 2015). In the case of fine mode absorbing aerosols overlying clouds, the absorption Ångström exponent leads to a greater impact on radiances reflected by the clouds at shorter wavelengths than longer ones (De Graaf et al., 2012; Torres et al., 2012). The "colour-ratio" approach has been applied to OMI (Ozone Monitoring Instrument - Torres et al., 2012) and MODIS (Moderate Resolution Imaging Spectroradiometer - Jethva et al., 2013) to simultaneously retrieve the aerosol and the cloud optical thicknesses over the SEAO. Using a similar technique, the MODIS retrieval developed by Meyer et al. (2015) takes advantage of the 6 channels of the instrument from the

UV to the Short-Wave Infra-Red (SWIR) to characterize not only the aerosol and cloud optical
thicknesses, but also the cloud droplet effective radius. For the first time, these studies have
provided large-scale observations of aerosols above clouds in the SEAO. However, these
approaches have been applied to satellite instruments on polar-orbiting platforms that provide
only two observations per day for MODIS (on the Aqua and Terra platforms) and one for OMI
and POLDER. The cloud cover over the SEAO has an important diurnal cycle which modulates
the DRE of aerosols during the day (Min and Zhang, 2014). Therefore, the study of the SEAO
cloud and above-cloud aerosol optical properties would benefit from the high temporal
resolution observations provided by geostationary satellite platforms.
Chang and Christopher (2016) have highlighted the ability of SEVIRI (Spinning Enhanced
Visible and Infrared Imager) to identify absorbing aerosols above clouds at high temporal
resolution. The instrument is on board the geostationary satellite MSG (Meteosat Second
Generation) and provides a full-disc observation every 15 minutes, offering a unique
opportunity to monitor the evolution of the cloud cover and to track aerosol plumes over the
SEAO. The objective of this two-part paper is to demonstrate the potential of this instrument
to retrieve simultaneously aerosol and cloud properties in the case of absorbing aerosols above
clouds. In this first contribution, we describe the approach used to derive the above-cloud
Aerosol Optical Thickness (AOT), the Cloud Optical Thickness (COT) and the Cloud droplet
Effective Radius (CER) and discuss the accuracy of the retrievals. The algorithm, as well as
the atmospheric correction scheme and the assumed aerosol model, are presented in Section 2.
The sensitivities in the retrieved quantities to the water vapour profile and the aerosol property
assumptions are assessed in Section 3. The evaluation of the stability of the retrieval is shown
in Section 4 and conclusions are drawn in Section 5. In a second companion paper, we will
compare our SEVIRI-based retrievals of cloud and aerosol properties with those from MODIS
products (Meyer et al., 2015) more comprehensively and also compare against *in situ* aircraft
observations from the CLARIFY-2017 field campaign.

## 2. Retrieval method

### a. Principle

The approach used to retrieve aerosol and cloud properties from satellite spectral radiance
measurements relies on the colour-ratio effect (Jethva et al., 2013). The signal backscattered
by a liquid cloud is almost spectrally neutral from the UV to the Near Infra-Red (NIR). On the
other hand, the absorption from biomass burning aerosols is typically larger at shorter
wavelengths. Therefore, the presence of absorbing aerosols above clouds modifies the apparent
colour of clouds. This enhancement of the spectral contrast can be detected by any passive
remote sensing instrument with two channels with enough separation in the UV/NIR region.
The SEVIRI instrument, aboard the MSG satellite (Aminou et al., 1997), has channels centred
at 0.64, in the visible, and at 0.81μm, in the NIR. Figure 1 plots the 0.81 μm radiance ($R_{0.81}$)
against the ratio of the 0.64 to 0.81 μm radiances ($R_{0.64}/R_{0.81}$), for absorbing aerosols above
clouds over an ocean surface for several aerosol and cloud optical thicknesses. Throughout this
paper, the radiances R refer to the normalized quantity as defined by Herman et al. (2005) and
the optical thicknesses (i.e. AOT, COT) are given at 0.55µm. The simulations have been
performed with the adding-doubling method (De Haan et al., 1987), considering a viewing
geometry of 20° for the solar zenith angle, 50° for the viewing zenith angle and 140° for the
relative azimuth. The cloud is located between 0 and 1 km and the aerosol layer is between 2
and 3 km. Aerosols have a refractive index of 1.54 - 0.025i and the size distribution follows a
lognormal with a geometric mean radius of 0.1µm. The cloud droplets have an effective radius
of 10 µm. Rayleigh scattering has been accounted for but the simulations do not include the
absorption from atmospheric gases. A Lambertian surface with an albedo of 0.05 is assumed.
For AOT = 0, the radiance ratio is around 1 and weakly depends on the COT. As the AOT
increases, the radiance at 0.81µm as well as the radiance ratio decreases, indicating that the
attenuation from the aerosol layer is larger at 0.64 µm. This attenuation is mainly due to the
absorption from the aerosol layer, which means that it is primarily correlated to the Absorption
AOT (AAOT).
As in the Nakajima and King technique (1990), the sensitivity of the retrieval to the CER is
brought by the Short-Wave Infra-Red (SWIR) channel of SEVIRI, centred at 1.64µm. Figure
2 shows the radiances at 0.81 and 1.64 µm for several COT and CER as well as the impact of
overlying absorbing aerosols. The simulations without aerosol are plotted in blue and represent
the signal typically used by cloud property retrievals that do not include light absorption from
overlying aerosols. The orange and red grids are associated with an AOT of 0.5 and 1.5 at
0.55µm. Compared to the no-aerosol case, these grids are shifted towards the upper left, which
means that the presence of aerosols decreases the NIR radiance and increases in the SWIR
signal. As highlighted by Haywood et al. (2004), not taking into account the aerosol absorption
above clouds leads to low biases in both the COT and the CER. These biases depend on the
aerosol loading as well as on the brightness of the underlying cloud.
Although the aerosol microphysical properties have some influence on the signal measured by
satellites, this kind of approach requires us to assume an aerosol model. Fundamentally, the
algorithm developed here aims to retrieve the above-cloud AOT, the COT and the CER from
the magnitude and the gradient of the radiances measured by SEVIRI at 0.64, 0.81 and 1.64
µm using a basic Look Up Table (LUT) approach and appropriate assumptions about the
aerosol model for the region (Haywood et al., 2003) that have been refined based on
measurements from the CLARIFY-2017 observational campaign (Zuidema et al., 2016).
**b. Atmospheric correction**
The SEVIRI channels chosen for the retrieval are fairly standard in atmospheric science and
have been widely used for aerosol and cloud analysis (e.g. Brindley and Ignatov, 2006;
Thieuleux, et al. 2005; Watts et al., 1998). However, the SEVIRI bandwidths are much larger
than other state-of-the-art instruments such as MODIS. Hence, SEVIRI radiances are
significantly more impacted by the absorption from various atmospheric gases. The spectral
response functions for the 0.64, 0.81 and 1.64 µm SEVIRI channels are plotted in Figure 3

together with the equivalent MODIS bands. The main absorbing gases in these spectral bands are ozone, water vapour, methane and carbon dioxide; gases which are typically produced and transported within biomass burning plumes (Browell et al., 1996; Koppmann et al., 2005). The contributions of each gas to the atmospheric absorption are also shown in Figure 3 and the two-way transmittances (i.e. from the top of the atmosphere to the cloud top and from the cloud top to the top of the atmosphere) weighted by the spectral response function have been calculated. For sake of simplicity, the two-way transmittances will be referred to as transmittances. Although the MODIS bandwidths are narrower than the SEVIRI ones, the weighted transmittances are similar for the 0.64 and 1.64 μm channels. In the NIR, the MODIS central wavelength (0.86 μm) is slightly larger than for SEVIRI (0.81 μm) and the spectral band is only weakly impacted by the humidity, with a weighted transmittance of 0.989. Within the SEVIRI band, water vapour absorption is much higher, with a transmittance of 0.931. As a result, humidity has an impact on the spectral contrast between the VIS and the NIR, and therefore, on the above-cloud AOT retrieval. The atmospheric correction, especially for the water vapour, is essential to accurately retrieve the aerosol and cloud properties from SEVIRI.

In order to correct the SEVIRI measurements for atmospheric absorption, the transmittances $T_{atm,\lambda}$ are calculated for each spectral band $\lambda$ from the cloud top height to the top of the atmosphere using the fast-radiative transfer model RTTOV (Matricardi et al., 2004; Hocking et al., 2014). The cloud top height is derived from the Met Office cloud property algorithm which uses the 10.8, 12.0 and 13.4 μm channels of SEVIRI (Francis et al., 2008, Hamann et al., 2014). Water vapour profiles come from the operational forecast configuration of the global Met Office Unified Model (Brown et al., 2012). This forecast is assimilated according to the scheme described by Clayton et al. (2013) that uses humidity data from various sources, including radiosondes and remote sensing sounding data from many meteorological satellites. The forecast is run every 6 hours and the humidity profile used for the atmospheric correction comes from the latest time-appropriate forecast field available. The profiles of the remaining gases - including ozone, carbon dioxide and methane - are those implicitly assumed by the RTTOV calculations (Matricardi, 2008). The radiance measured by SEVIRI $R_{atm,\lambda}$ is finally corrected using:

$$R_{atm,\lambda} = T_{atm,\lambda}R_\lambda \qquad (1)$$

where $R_\lambda$ is the radiance corrected from the gaseous absorption.

### c. Aerosol model

The choice of the aerosol microphysical properties to use for the retrieval is similar to that of Haywood et al (2003), but based on more comprehensive *in situ* measurements acquired during the CLARIFY-2017 field campaign. The Facility for Airborne Atmospheric Measurements (FAAM) BAe 146 aircraft was deployed in August-September 2017 operating from Ascension Island, with a main objective of studying biomass burning aerosol interactions with both radiation and clouds over the SEAO. This analysis focuses on flight C050, performed on 04 September, 2017. A profile descent from 7.3 to 1.9 km altitude was performed in order to sample the aerosol layer above clouds.


The aerosol dry extinction and absorption were measured with the EXSCALABAR instrument
(EXtinction, SCattering and Absorption of Light for AirBorne Aerosol Research), which
consists of a series of cavity ring-down and photoacoustic absorption cells operating at
different wavelengths (Davies et al., 2018). From these *in situ* measurements, the Single
Scattering Albedo (SSA) has been calculated at the instrument wavelengths of 405 and 658
nm. The uncertainty in SSA calculations are related to the corresponding uncertainties in the
extinction and absorption coefficients measured by EXSCALABAR. This error analysis has
been performed previously and the reader is directed to Davies et al. (2019). Briefly, the
measured extinction has an accuracy of ~2%, and we use a 2% extinction uncertainty in the
analysis here. The errors in absorption measurements using photoacoustic spectroscopy depend
on uncertainties in the ozone calibration, microphone pressure dependence and the background
response from laser scattering/absorption on the windows of the photoacoustic cell. We have
shown in recent publications that our calibration uncertainties are ~5% (Cotterell et al. 2019;
Davies et al. 2018), and the uncertainty in the pressure-dependent microphone response is 1.2%
(Davies et al. 2019). The background response from laser-window interactions is from 0.27
and 0.54 Mm$^{-1}$. Thus, the total absorption uncertainty, propagating all the above uncertainties,
is absorption-dependent and ranges from 29.0 – 55.0 % (dependent on PAS measurement
wavelength) at 1 Mm$^{-1}$ and 8.1 % at 100 Mm$^{-1}$ (independent of PAS measurement wavelength).
We propagated these total measurement uncertainties for both extinction and absorption
measurements to derive the standard deviation $\sigma$ in our calculated SSA values. We find that
the mean SSA uncertainties are 0.013 and 0.018 at the measurement wavelengths of 405 and
658 nm respectively.

The aerosol size distribution was characterized between 0.05 and 1.50 μm radius using a wing-
mounted Passive Cavity Aerosol Spectrometer Probe (PCASP). Before and after the campaign,
the bin sizes of the PCASP were calibrated using aerosolized diethyhexyl sebacate and
polystyrene latex of known size and refractive index (Rosenberg et al., 2012). Further Mie-
scattering theory based calculations are performed in order to determine the bin sizes at the
refractive index of the biomass burning aerosol sample. Partial evaporation of water is expected
in the PCASP due to the heating of the probe, which may decrease the aerosol size. However,
the sonde dropped during the flight indicates an average relative humidity above clouds of
29.2% with a maximum of 38.6%. According to Magi and Hobbs (2003), the light scattering
coefficient of an aged African biomass burning plume only increases by a factor of 1.01 for a
relative humidity of 40%. For this reason, the impact of humidity on the PCASP and
EXSCALABAR measurements is neglected. Three sources of errors have been taken into
account on the PCASP measurements: the error on the bin concentration is calculated
according to Poisson counting statistics, the sample flow rate error is assumed to be 10% and
a bin edge calibration error of half a bin has been considered.

The aerosol properties needed for the SEVIRI retrieval include the size distribution and the
complex refractive index. The normalized number size distribution (dN/dlnr) is commonly
represented by a combination of lognormal modes:

$$\frac{d\,N}{d\ln r} = \sum_i \frac{N_i}{\sqrt{2\pi}}\,\frac{1}{\ln\sigma_i}\,exp\left[\frac{-(\ln r_i - \ln r)^2}{2(\ln\sigma_i)^2}\right] \qquad (2)$$

where $N_i$, $r_i$ and $\sigma_i$ are the number fraction, the geometric mean radii and the standard deviation of the mode $i$, respectively. As in most remote sensing applications, it has been chosen to represent the particle size distribution for the aerosol during CLARIFY-2017 with a fine and a coarse mode contributions. The aerosol optical properties are calculated using the Mie theory, as the spherical approximation is expected to be valid for biomass burning particles from one hour after being released in the atmosphere (Martins et al., 1998). The aerosol model is selected by iteratively adjusting the refractive index and fitting the PCASP measurements (Fig. 4a) until the aerosol model matches the SSA from EXSCALABAR (Fig. 4b). In order to obtain the most suitable aerosol optical parameters for the retrieval, it is important to accurately fit the PCASP measurements where the aerosols contribute the most to the SEVIRI signal. Each bin of the PCASP has been assigned a weight for the fit of the bimodal distribution. The weights have been calculated in a similar way to Haywood et al. (2003), which means that they are proportional to the contribution of each bin to the total aerosol extinction in the 0.6 μm band. The bins corresponding to the 0.15 to 0.25 μm radius range contribute to about 77% of the extinction. Consequently, these bins have been assigned appropriate larger weights during the fitting process of the size distribution. Due to the small fraction of coarse mode aerosols, the standard deviation of this mode $\sigma_{coarse}$ could not be reliably fitted and has been set to a value of 2.23, which is within the same order of magnitude than the one assumed for absorbing aerosol (~2.12) in the MODIS Dark Target operational algorithm (Levy et al., 2009).

The aerosol model that best represents the PCASP and EXSCALABAR measurements is shown in blue on Figures 4a and 4b. A refractive index of 1.51-0.029i has been obtained, associated with an SSA of 0.85 at 0.55 μm which is within the range of SSA measured over the SEAO during the SAFARI and the DABEX campaigns (Johnson et al., 2008) and on the upper end of the values from Ascension Island reported by Zuidema et al. (2018). Regarding the refractive index, it should be noted that the SSA is not very sensitive to the real part suggesting that the value of 1.51 is not particularly well constrained. However, a real part of 1.51 is consistent with the AERONET retrievals for African biomass burning particles (Sayer et al., 2014) and is adopted here. The best-fit size distribution is characterised by [$r_{fine}$, $\sigma_{fine}$, $N_{fine}$; $r_{coarse}$, $\sigma_{coarse}$, $N_{coarse}$] = [0.12μm, 1.42, 0.9996; 0.62μm, 2.23, 0.0004]. By way of comparison, the 3-mode lognormal distribution obtained for aged biomass burning aerosols during the SAFARI 2000 campaign (Haywood et al., 2003), defined by [$r_1$, $\sigma_1$, $N_1$; $r_2$, $\sigma_2$, $N_2$; $r_3$, $\sigma_3$, $N_3$] = [0.12μm, 1.30, 0.996; 0.26μm, 1.50, 0.0033; 0.80μm, 1.90, 0.0007], is plotted in orange on Figure 4a. The radius associated with the first mode is consistent with the CLARIFY-2017 model. The absence of the second fine mode in this study is compensated by a larger standard deviation for the fine mode. Finally, the radius of the CLARIFY-2017 coarse mode is slightly smaller than the SAFARI-2000 one but the coarse mode fractions of the two models are close to each other. The uncertainties on the aerosol properties have been estimated using the errors on the PCASP and EXSCALABAR measurements. The uncertainty on the imaginary part of the refractive index is 0.02 for the real part and 0.004 for the imaginary part. For the

size distribution, the uncertainty is 0.016μm, 0.09 and 0.00045 for radius, the standard
deviation and the number fraction of the fine mode respectively.

**d. Algorithm**

The algorithm relies on the comparison of the corrected SEVIRI signal at 0.64, 0.81 and 1.64
μm with precomputed radiances. The simulations have been performed using an adding-
doubling radiative transfer code (De Haan et al., 1987). The surface is assumed to be
Lambertian with an albedo of 0.05 at all wavelengths which is typical of the sea-surface albedo
under diffuse radiation conditions. The aerosol and cloud properties assumed for the LUT are
summarized in Table 1. The truncation of the cloud droplet phase function has been done using
the delta-M method (Wiscombe, 1977) and the TMS correction (Nakajima and Tanaka, 1988)
has been applied. The cloud layer is assumed to be located between 0 and 1 km and the aerosol
layer between 2 and 3 km. The sensitivity of the algorithm to the altitudes of the aerosol and
cloud layers is expected to be negligible due to the small contribution of the Rayleigh scattering
to the signal at the SEVIRI wavelengths. We have evaluated the error due to the fixed aerosol
and cloud altitudes to be lower than 2.5% on the AOT and 0.3% on the cloud properties. The
cloud droplets are assumed to follow a gamma law distribution characterised by an effective
variance of 0.06. When the cloud is optically thin and/or the cloud droplets are too small, it is
not possible to separate the contribution to the optical signal arising from aerosols from that of
clouds. Therefore, the minimum values for the CER and the COT in the LUT are 4 μm and 3,
respectively. This also justifies the assumption of a relatively simple sea-surface reflectance
parameterisation as, at COTs exceeding 3, the sea-surface has little impact on the upwelling
radiances above clouds. Clouds associated with lower COT and/or CER are rejected. The
aerosol model corresponds to the CLARIFY-2017 model mentioned above, assuming the same
refractive index at the 3 SEVIRI wavelengths.

The retrieval of the above-cloud AOT, COT and CER is performed simultaneously. The result
corresponds to the parameters that minimise the difference $\varepsilon$ between the simulated radiances
$R_{sim}$ and the corrected satellite signal $R_\lambda$:

$$\varepsilon = \sum_\lambda \left(\frac{R_\lambda - R_{sim,\lambda}}{R_\lambda}\right)^2 \tag{3}$$

When the simulated signal is not close enough to the satellite measurements (i.e. $\varepsilon > 0.0006$),
the result is rejected. The retrieval of the above-cloud AOT is highly uncertain at the cloud
edges and for inhomogeneous clouds. In order to remove these results, the products are
aggregated onto a 0.1 × 0.1° grid and the standard deviation of the AOT and the CER are
calculated. Note that each grid cell represents approximately 12 SEVIRI pixels. The
inhomogeneity parameter $\rho$ is defined by the ratio of the standard deviation of a parameter to
the average value of this parameter. The results corresponding to a standard deviation of the
AOT larger than 0.7 and/or $\rho_{CER} > 0.2$ as well as grid cells associated with less than 9 successful
retrievals are rejected.

It is important to realise that the uncertainties that we quantify here are structural and
parametric uncertainties related to assumptions made in the retrieval algorithm. When using a
fixed aerosol model, no account is made for natural variability in the aerosol optical parameters
and the associated uncertainty; this is dealt with in the uncertainty analysis that follows.

## 3. Results and uncertainty analysis


### a. Case study


The algorithm has been applied to an event of biomass burning aerosols above clouds captured
by SEVIRI on 28 August 2017 at 10:12 UTC. The RGB composite, the retrieved above-cloud
AOT, COT and CER over the SEAO region are shown in Figure 5. The largest AOT are
observed off the coast of Angola, with a local average value of 1.0 and a maximum of 1.6 at
0.55 μm. The AERONET site of Lubango (14.96 ˚S - 13.45 ˚E) measured an average AOT of
0.75 that day with an Ångström exponent of 1.83, indicating the expected domination of fine
mode biomass burning aerosols. A gradient of AOT is observed towards the south-west, as we
move away from the source as might be expected from a pre-campaign analysis of satellite
retrievals (Zuidema et al., 2016). Absorbing aerosols above clouds are also detected in the
north-west part of the region. Around Ascension Island (7.98 ˚S - 14.42 ˚W), the above-cloud
AOT from SEVIRI is around 0.37 while the AERONET site indicates a value of 0.48 associated
with an Ångström exponent of 1.271. This suggests that coarse mode aerosols, such as sea salt
within the boundary layer but generally below cloud, are contributing to the total column
aerosol load. The cloud properties retrieved are within the range of values typically observed
for marine stratocumulus (Szczodrak et al., 2001) with more than 90 % of the COT lower than
25 and 99 % of the CER between 4 and 20 μm. As a comparison, Figure 6 shows the equivalent
aerosol and cloud properties retrieved from MODIS-Terra with the MOD06ACAERO
algorithm (Meyer et al., 2015) for the 10:00 and 11:30 UTC overpasses. The MODIS above-
cloud AOT pixels associated with an uncertainty larger than 100% have been removed. A good
spatial agreement is observed between the two satellites products. The above-cloud AOT from
MODIS is also 1.0 on average close to the coast. On average over the area, the MODIS above-
cloud AOT is larger by 0.05 compared to SEVIRI. Considering that MODIS is less sensitive
to the atmospheric absorption and that the two algorithms are based on the same principle, the
small differences observed between the two above-cloud AOT tend to validate the atmospheric
correction applied on the SEVIRI measurements for that case.  There is a good consistency
between the MODIS and the SEVIRI COT. Finally, the CER retrieved with the
MOD06ACAERO algorithm is larger by 2.2 μm compared to the SEVIRI CER. This almost
systematic difference is mainly due to differences in the satellite instruments, and especially,
the difference in the channels used for the retrieval (Platnick, 2000). A fully statistical analysis
against the MODIS algorithm, and against airborne remote sensing and *in situ* measurements
will be presented in a companion paper.

### b. Atmospheric correction

The atmospheric transmittances above clouds used to correct the SEVIRI measurements from the gas absorption are calculated based on forecast water vapour profiles. In order to assess the sensitivity of the retrieval to the atmospheric correction, new transmittances have been calculated for the event studied here, modifying the specific humidity by +/-10%. The aerosol and cloud properties retrieved with the modified atmospheric corrections are aggregated on a $0.1 \times 0.1°$ grid. Figure 7 compares the retrieved aerosol and cloud properties from SEVIRI-measured radiances using the original specific humidity forecast with the perturbed specific humidity (+10% in orange and -10% in blue). The uncertainty on the water vapour content impacts mainly the retrieval of the above-cloud AOT, and then the COT, because of its effect on the radiance ratio. A +10%/-10% bias on the humidity leads to an overestimation/underestimation of the AOT and COT respectively. On average, errors of 18.5%, 5.5% and 2.3% have been calculated for the AOT, COT and CER respectively, based on biases of 10% in the specific humidity forecast. These errors are likely upper estimates because forecast errors in specific humidity are unlikely to reach these values owing to the extensive assimilation of satellite data and sonde profiles by the data assimilation process used in the Met Office forecast model as previously mentioned. However, the differences between forecast model specific humidities and those of simple standard atmosphere climatological values (e.g. those of McClatchey et al., 1972) frequently exceed 10%, indicating that accurate retrievals of aerosol and cloud need synergistic retrievals or data assimilated forecasts of specific humidity.

### c. Aerosol model

The LUT used for the SEVIRI retrieval uses an assumed aerosol model based on *in situ* measurements from CLARIFY-2017. However, the absorption property and the size of biomass burning particles are expected to vary during the fire season and across the SEAO (e.g. Eck et al., 2003). Here, we analyse the impact of the aerosol assumptions on the retrieved aerosol and cloud properties.

In order to create a range of aerosol optical properties, a thousand aerosol models have been processed using the Mie theory. The radius and the standard deviation of the fine mode, and the real and imaginary part of the refractive index of the models are random values following a normal distribution. Their mean corresponds to the CLARIFY model values provided in Table 1, with standard deviations of 0.01μm and 0.1 for the radius and the standard deviation of the fine mode, 0.02 for the real part of the refractive index and 0.008 for the imaginary part. Figure 8a and 8b show the histograms of the simulated SSA and asymmetry factor g at 0.55 μm in orange. As a comparison, histograms of the AERONET SSA and g are plotted in blue. The data corresponds to the AERONET level 2.0 retrievals for August-September, from 1997 to 2018 and for inland sites of Southern Africa (10°S–35°S, 10°E–40°E). Only data associated with an Ångström exponent larger than 1.0 have been used in order to remove measurements dominated by coarse mode particles (such as dust and sea salt) that are less likely to be observed

above clouds in the SEAO. The mean SSA (0.862) and the mean g (0.620) from AERONET
are respectively slightly larger and smaller than the CLARIFY model. Small differences
between above-cloud and full column aerosol properties could be explained by the contribution
of aerosol within the boundary layer, such as pollution, desert dust and sea salt. The dashed
lines in Figure 8a and 8b represent the mean +/- the standard deviation of SSA and g. The
AERONET standard deviation is 0.023 for the SSA and 0.024 for g while the simulation
produces a standard deviation of 0.036 for the SSA and 0.041 for g. The simulated range of
both optical properties is larger than the range observed by AERONET. Therefore, the
variation of the aerosol microphysical properties used for the simulations is wide enough to
cover the range of observed aerosol optical properties.

From the simulated standard deviation $\sigma$ of g and SSA, eight aerosol models have been defined
and their properties are summarized in Table 2. The first four are used to test the sensitivity of
the retrieval to g and SSA independently ([$SSA_{CLARIFY}$+/-$\sigma_{SSA}$, $g_{CLARIFY}$] and [$SSA_{CLARIFY}$,
$g_{CLARIFY}$+/-$\sigma_g$]) and the sensitivity to both parameters will be assessed with the last four
([$SSA_{CLARIFY}$+/-$\sigma_{SSA}$, $g_{CLARIFY}$+/-$\sigma_g$]). New LUTs have been processed with these modified
aerosol models and used to re-process the case study from section 3.a. After aggregating the
data on a 0.1 × 0.1° grid, the AOT as well as the Absorption AOT (AAOT), the COT and the
CER are compared against those obtained with the standard CLARIFY-2017 aerosol model.
Results are shown in Figure 9 and 10. For each aerosol and cloud property, a linear relationship
is observed between the retrieval using the standard CLARIFY-2017 aerosol model and the
modified one. The retrieval of cloud properties (fig. 9c, 9d, 10c and 10d) appears to be weakly
sensitive to the assumed aerosol model, with g having a slightly larger impact. On average,
differences lower than 4.1% are observed on the COT and lower than 2.4% on the CER. As
expected, the choice of the aerosol model has much more influence on the AOT retrieval. The
uncertainty on the AOT is dominated by the SSA assumption. When aerosols are more
absorbing than the CLARIFY model, the algorithm overestimates the AOT by 25.7%.
Conversely, the retrieved AOT is underestimated by 32.6% when aerosols are less absorbing
than the CLARIFY model. The impact of g alone on the retrieved AOT is far less significant
and lower than 4.3%. Figure 9a, which shows the impact of a perturbation on both the SSA and
g, confirms that the SSA is the parameter with the strongest influence on the AOT retrieval.
The largest overestimation (27.5%) is observed when both the SSA and g are overestimated
(fig. 10a), while the largest underestimation (-33.3%) is obtained when the SSA is
underestimated and g is overestimated. The retrieval of the above-cloud AOT depends mostly
on the aerosol absorption of the light reflected by the cloud. Therefore, it is expected that the
retrieved AAOT is less sensitive to the absorbing property of the aerosol than the AOT. The
sensitivity of the AAOT to the assumed aerosol properties is shown in Figure 9b and 10b. The
uncertainty in the AAOT due to an error in g is similar to the uncertainty in the AOT (<5%).
However, the influence of the SSA assumption alone on the AAOT is smaller than the influence
on the AOT, with differences of 1.9% and -8.7%. This means that a perturbation of the SSA
primarily impacts the scattering AOT. The largest overestimation of the AAOT (2.7%) is
obtained when the assumed aerosol model overestimates g. An underestimation of the SSA and
an overestimation of g lead to the largest underestimation of the AAOT (-5.1%).

The variation of the solar zenith angle, and therefore, of the satellite observation geometry
during the day can impact the sensitivity of the retrieval to the aerosol assumptions. Therefore,
the 15-minute SEVIRI observations for the 28 August have been processed using the eight
aerosol models described above and compared to the aerosol and cloud properties retrieved
with the CLARIFY aerosol model. The difference $\Delta x_i$ of a product x is defined as:
$$\Delta x_i = (x_{CLARIFY} - x_i)/x_i \times 100\%$$
where $x_{CLARIFY}$ and $x_i$ is the mean product x retrieved over the SEVIRI slot with the aerosol
CLARIFY model and the modified model i, respectively. Figure 11 shows the time series of
$\Delta$AOT (a), $\Delta$AAOT (b), $\Delta$COT (c) and $\Delta$CER (d) obtained with the modified aerosol models.
The sensitivity of the retrieved cloud properties to the aerosol model assumptions remains
small (lower than 5.6% for the COT and 2.6% for the CER) and dominated by the sensitivity
to g. Apart from a small decrease of $\Delta$COT at midday when g is overestimated (solid blue line)
and an increase of $\Delta$COT in late afternoon when the SSA is underestimated (solid red line), no
significant trend is observed on the cloud property sensitivities. As observed previously, the
uncertainty on the AOT is led by the SSA assumption, with the AOT being overestimated
(respectively underestimated) when the assumed SSA is overestimated (respectively
underestimated). Until 15:00, $\Delta$AOT stays within +/-40%, with the sensitivity to the SSA being
slightly larger at midday. Then it increases up to 60% when the SSA is overestimated and g is
underestimated (dashed blue line). Similar trends are observed on $\Delta$AAOT, with generally
lower values than $\Delta$AOT. An increase of the uncertainty is observed on the AAOT after 15:00,
that reaches up to 27% at 16:30. Before 15:00, there is a larger AAOT sensitivity to the SSA
around midday (+8.9%/-15.2%), but there is no evident evolution of the sensitivity to g with
time. The case that lead to the largest biases on the AAOT is when the SSA is underestimated
and g overestimated (dashed green lines), with an underestimation of up to 23%. However, it
should be noted that 0% of the AERONET observations used in Figure 8 are associated with
an SSA lower than $SSA_{CLARIFY}-\sigma_{SSA}$ and a g larger than $g_{CLARIFY}-\sigma_g$. Otherwise, the sensitivity
of the AAOT to the aerosol property assumptions stays between -16.6 and +9% before 15:00.

In conclusion, the retrieved AOT is less sensitive to the aerosol property assumption before
15:00, with an uncertainty of 40%. This uncertainty is dominated by the sensitivity of the
retrieval to the SSA. An overestimation (respectively underestimation) of the AOT is expected
when the observed aerosols are more (respectively less) absorbing than the aerosol model
assumed for the retrieval. A better accuracy is obtained on the retrieved AAOT, with an
uncertainty generally lower than 17 % before 15:00. The sensitivity of the cloud properties to
the aerosol model assumption remain small all day long, with an uncertainty of 5.6% on the
COT and 2.6% on the CER.

**4. Assessing the stability of the retrieval**

One of the major benefits from using SEVIRI is the ability to track both aerosol and cloud
events at high temporal resolution. Therefore, it is important to evaluate how consistent the
retrieval is over time. For that purpose, two days of continuous observations (i.e. 5[th] and 6[th]
September 2017) have been analysed and the retrieved properties have been averaged over
20˚S and 10˚S, and 5˚E and 15˚E, which correspond to the red square on the maps of Figure
12. The above-cloud AOT, COT and CER time series are presented in Figures 13a, b and c.
The studied area is located next to the coast, where the AOT is typically the highest. The above-
cloud AOT is around 0.66 and 0.72 for the $5^{th}$ and the $6^{th}$ September, respectively. As expected,
the transport of the aerosol plume from east to west is slow, resulting in a small evolution of
the above-cloud AOT. On both days, a peak is observed at 12:12pm with an anomaly larger
than the AOT variability. This localised discontinuity in the above-cloud AOT is shown in the
11:42, 12:12 and 12:42 UTC maps for 05 September 2017 of Figure 12. The evolution of the
cloud properties is slightly more complex. A small decrease is observed on both the COT and
CER until 2pm. After 3pm, both properties sharply increase. The clouds are strongly affected
by the diurnal cycle and a shoaling of the cloud cover is expected from early morning to late
afternoon. As the thinnest clouds vanish, the cloud fraction decreases together with the number
of retrievals in the area. This results in a larger contribution of the thickest clouds to the mean
value in the late afternoon. As for the above-cloud AOT, large variations of the CER are
observed around noon. At that time, the sun and the satellite are almost aligned and the
scattering angle (fig. 13d) reaches values larger than 175˚ which corresponds to the region
where the glory phenomenon is typically observed. Several reasons can explain why the
retrieval does not perform well in backscattering direction. The first one is the uncertainty in
the LUT due to the truncation of the cloud phase function. Although the TMS correction gives
good results, biases still remain in the glory aureole (Iwabushi and Suzuki, 2009). Also, the
radiances in the glory are more sensitive to the cloud droplet microphysics (Mayer et al., 2004).
The assumption on the variance of the droplet size distribution may induce biases in the
retrieval. Therefore, the accuracy of the retrieval cannot be guaranteed within the glory aureole
and these observations should be discarded. In Figure 13, the timespans corresponding to the
MODIS Aqua and Terra overpasses in the region are highlighted in orange. This shows that
MODIS measurements are typically performed before and after SEVIRI observes the glory
backscattering over the SEAO, usually allowing comparisons between these instruments.
The performance of the algorithm is further assessed by evaluating the stability of the retrieved
above-cloud AOT at pixel level. As noted by Chang and Christopher (2016), in this region over
these scales, aerosols are expected to have a limited temporal variability and the variation of
the above-cloud AOT is expected to be small between t=0 and t+/-15 minutes. The differences
between the AOT retrieved at t=0 and the running mean estimated between t-15 and t+15
minutes have been calculated at pixel level for observations between 09:00-15:00 UTC,
removing measurements within the glory backscattering region. Figure 14 shows the histogram
of the AOT differences calculated over a 12-day period (01 to 12 September 2017). The
differences follow a normal distribution centred around 0.0 with a standard deviation of 0.1.
This short-term variability can be attributed to several sources of uncertainties, such as the total
amount of water vapour, its vertical distribution, the retrieved cloud top height and the
numerical fitting procedure. This analysis indicates that the retrieval of the above-cloud AOT
remains relatively stable, with an observed variability of +/-0.1 between consecutive
observations. Except for the glory backscattering, the stability observed on the retrieved aerosol
and cloud properties reinforces the reliability of the algorithm.

## 5. Conclusion

Recently, progress has been made in the remote sensing field in order to fill the lack of aerosol above cloud observations. Techniques have been developed to retrieve aerosol and cloud properties over the SEAO from passive remote sensing instruments. These algorithms take advantage of the colour-ratio effect (Jethva et al., 2013), which is the spectral contrast produced by the aerosol absorption above clouds. Although OMI (Torres et al., 2012), MODIS (Jethva et al., 2013; Meyer et al., 2015) and POLDER (Peers et al., 2015) already provide useful information about aerosols above clouds, these instruments are on polar-orbiting satellites and their low temporal resolutions prevent monitoring the diurnal variation of the cloud cover and of the DRE of aerosols over the SEAO. For the first time, we have applied a similar algorithm to geostationary measurements from the SEVIRI instrument, which has a repeat cycle of 15 minutes. The method consists of a LUT approach, using the channels at 0.64, 0.81 and 1.64 μm in order to retrieve simultaneously the above-cloud AOT, COT and CER.

Compared to other satellite instruments, the SEVIRI measurements are more sensitive to the absorption from atmospheric gases because of their wider spectral bands. Therefore, an efficient atmospheric correction scheme is essential in order to separate the absorption from aerosol absorption and from the atmospheric. Atmospheric transmittances are calculated with the fast-radiative transfer model RTTOV based on the cloud top height observed by SEVIRI and the forecasted water vapour profiles from the Met Office Unified Model. The water vapour correction has the largest impact on the above-cloud aerosol retrieval. The impact of errors in the atmospheric correction has been evaluated by modulating the humidity profile for a case study. A positive bias of both the AOT and the COT is observed when the water vapour is overestimated, and vice versa. On average, an 18.5% bias on the AOT and a 5.5% bias on the COT are expected for a 10% error on the water vapour profile. Although a good accuracy is expected from the forecast model, this limitation should be kept in mind when utilising or further developing SEVIRI products. In the companion paper, the humidity from the forecast will be compared against the dropsonde measurements from the CLARIFY-2017 campaign.

The choice of the aerosol model used to produce the LUT is also a key feature of the method. *In situ* measurements of aerosols above clouds have been performed off the coast of Ascension Island during the CLARIFY-2017 field campaign. An aerosol model optimised for the SEVIRI spectral bands has been obtained by analysing the vertical profiles of extinction and absorption from EXSCALABAR together with the size distribution from a PCASP. A bimodal lognormal distribution has shown to adequately reproduce the observations. A fine mode radius of 0.12 μm has been obtained, which is in good agreement with the biomass burning measured over the SEAO during SAFARI 2000 (Haywood et al., 2003). The refractive index has been evaluated at 1.51-0.029i. The corresponding SSA of 0.85 at 0.55 μm is consistent with both *in situ* and remote sensing observations of African biomass burning aerosols (Johnson et al., 2008; Sayer et al., 2014). In addition to the uncertainty associated with the estimation of the aerosol model, a seasonal dependence is expected in the biomass burning properties as well as

modifications due to aging processes during their transport over the SEAO. We have evaluated the impact of applying a single model assumption on both aerosol and cloud properties. Retrievals have been performed considering aerosol models with modified SSA and asymmetry factor g. It has been shown that the sensitivity of the retrieved cloud properties to the aerosol model assumption is small with errors lower than 5.6% on the COT and 2.6% on the CER. As expected the impact of the assumed aerosol properties is much larger on the above cloud AOT, with an uncertainty estimated at 40% before 15:00 UTC. This uncertainty is led by the sensitivity of the retrieval to the SSA. Because the method relies on the impact of the aerosol absorption on the light reflected by the clouds, the perturbation of the SSA has primarily an impact on the scattering contribution of the AOT. Therefore, a better accuracy is obtained on the retrieved AAOT, with biases generally lower than 17% before 15:00 UTC. After that time, an increase of the uncertainty on both the AOT and the AAOT has been observed, and users are advised to be careful when using the late afternoon aerosol product. For any satellite retrievals based on the colour-ratio technique, aerosol properties, including the SSA, have to be assumed and the same order of magnitude can be expected on the sensitivity of their AOT. This analysis highlights the importance of a suitable constrain on the SSA.

Despite the wider channels and the narrower spectral range of SEVIRI, it has been demonstrated that the geostationary instrument has the potential to detect and quantify the absorbing aerosol plumes transported above the clouds of the SEAO. Except from observations within the glory backscattering for which the retrieval has shown to be unstable, a good consistency has been observed on the aerosol and cloud properties. The stability of the results during the day is promising for future uses of the SEVIRI algorithm. In the companion paper, the reliability of the retrieved aerosol and cloud properties will be further assessed by analysing the consistency with the MODIS retrievals and comparing with direct measurements from the CLARIFY-2017 field campaign. The potential of such a retrieval is obvious. The 15-minute resolution will aid in tracking the fate of above-cloud biomass burning aerosol and will prove invaluable for assessing models of the emission, transport and deposition of biomass burning aerosol, with implications for accurate determination of the direct radiative effects of biomass burning aerosol at high temporal resolution.

## **Author contribution**

FP, PF and JMH developed the concept and the ideas for the conduction of this paper. PF implemented the atmospheric correction scheme and FP, the retrieval algorithm. CF, SJA, KS, MIC, NWD and JMH operated, calibrated and prepared the *in situ* measurements from EXSCALABAR and the PCASP. The reliability of the retrieved products was analysed throughout the development of the algorithm with the help of KGM and SEP. FP carried out the analysis and prepared the manuscript with contributions from all co-authors.

## **Acknowledgement**

This research was funded by the NERC CLARIFY project NE/L013479/1. Further support was provided by the Research Council of Norway via the projects AC/BC (240372) and NetBC (244141)

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

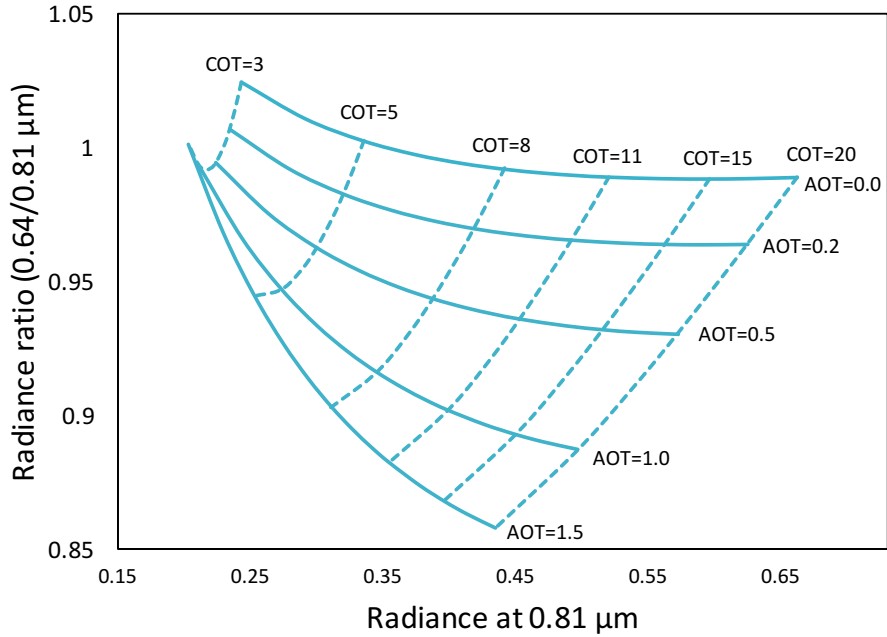

**Figure 1:** Radiance ratio $R_{0.64}/R_{0.81}$ as a function of the radiance at 0.81μm for absorbing
aerosols above clouds simulated with the adding-doubling method (De Haan et al., 1987).
COTs and AOTs are indicated at 0.55 μm.

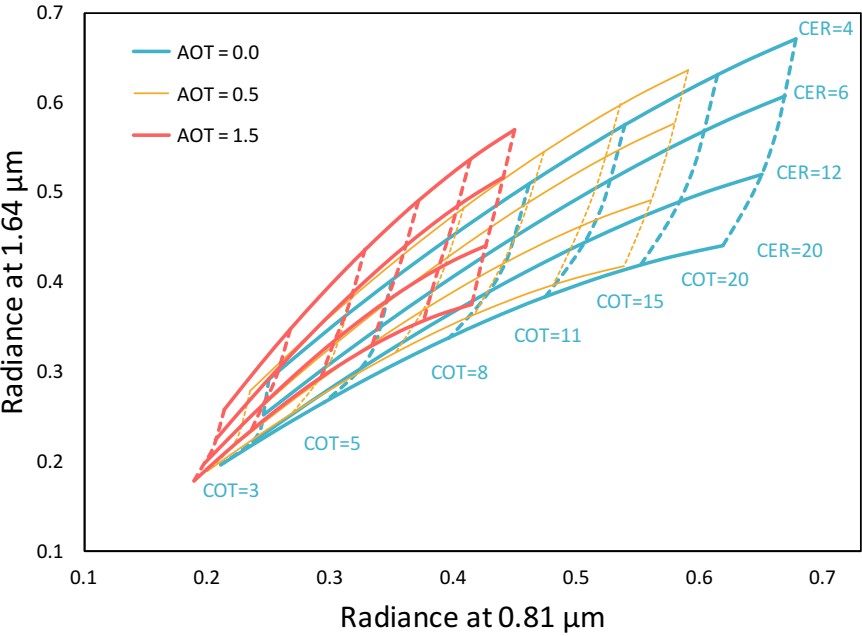

**Figure 2:** Simulated radiances at 1.64 and 0.81μm for clouds with varying COTs and CERs
(in μm), without (blue) and with (orange and red) overlying absorbing aerosols above. The
viewing geometry, the aerosol and the cloud properties are the same as Figure 1.

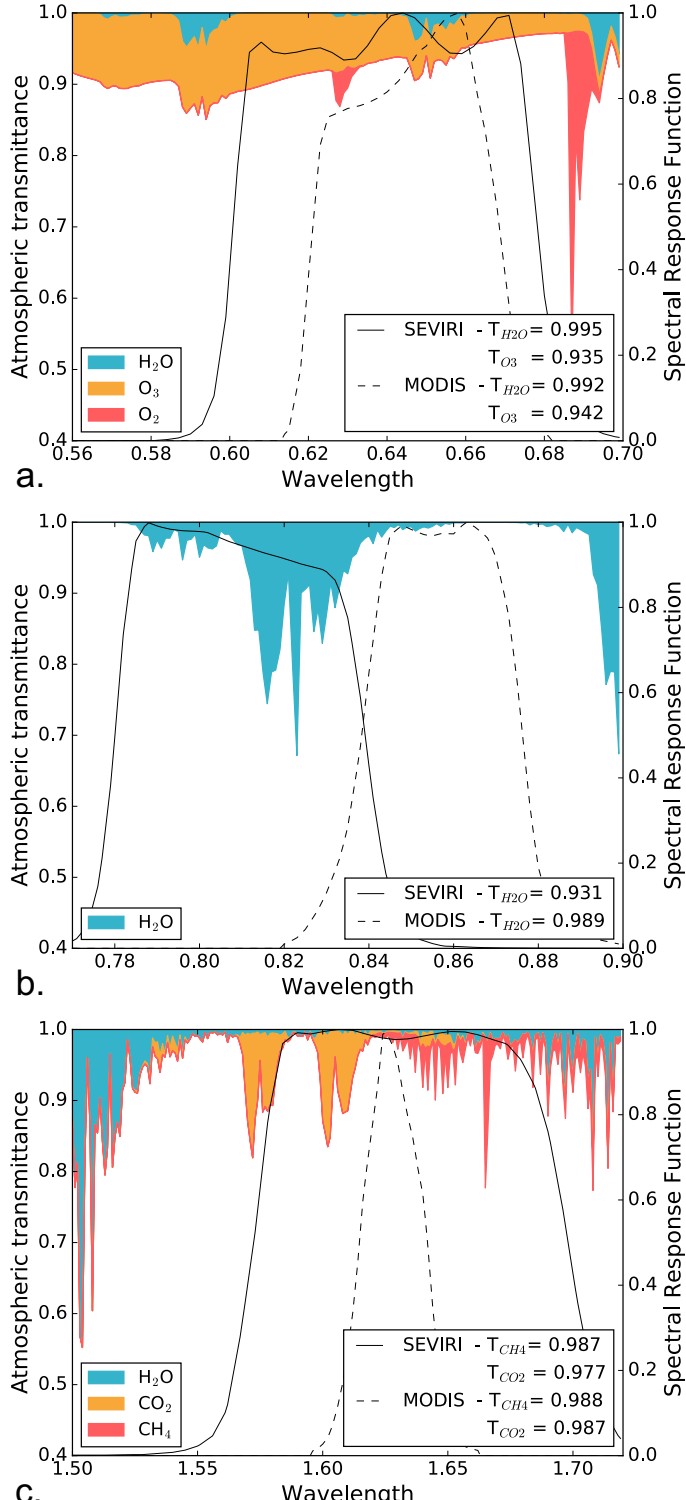

**Figure 3:** Spectral response function of the SEVIRI bands at 0.64 (a), 0.81 (b) and 1.64 μm (c)
with the corresponding MODIS ones (dashed lines) as well as the atmospheric transmittance
within the spectral range (in colour). The transmittances have been calculated with the
SOCRATES radiative transfer scheme (Manners et al., 2015; Edwards and Slingo, 1996)
assuming a humidity profile measured during SAFARI (Keil and Haywood, 2003). In the
legend of each plot, the transmittance weighted by the spectral response function is given for
the main absorbing gases.

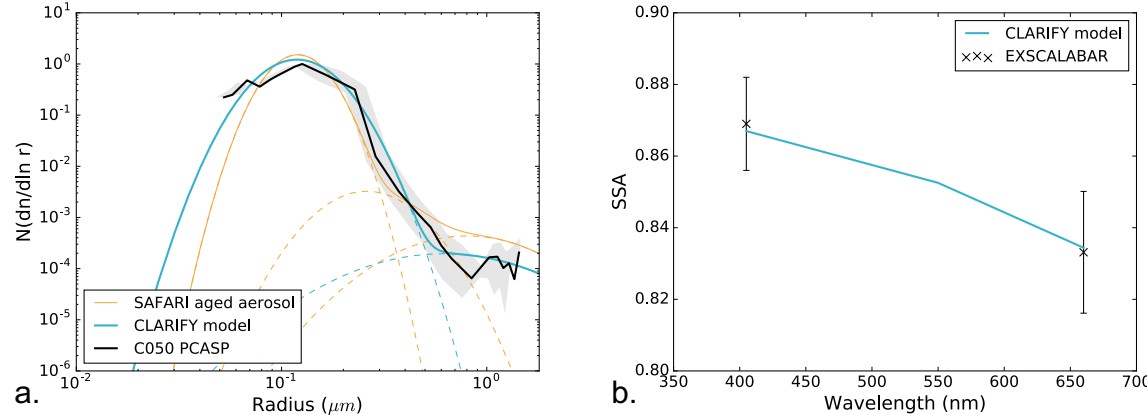

**Figure 4:** Normalized size distribution (a) and SSA (b) measured above clouds during flight
C050 of the CLARIFY-2017 campaign (black). The grey shade area represents the PCASP
measurement and calibration uncertainties. Blue lines represent the fitted aerosol model, the
orange lines correspond to the aged aerosol size distribution from SAFARI (Haywood et al.,
2003), and the dashed lines shows the contribution of each mode. CLARIFY-2017 aerosol
model: [$r_{fine}$, $\sigma_{fine}$, $N_{fine}$; $r_{coarse}$, $\sigma_{coarse}$, $N_{coarse}$] = [0.12µm, 1.42, 0.9996; 0.62µm, 2.23, 0.0004],
refractive index = 1.51 – 0.029i. SAFARI aged aerosol model: [$r_1$, $\sigma_1$, $N_1$; $r_2$, $\sigma_2$, $N_2$; $r_3$, $\sigma_3$, $N_3$]
= [0.12µm, 1.30, 0.996; 0.26µm, 1.50, 0.0033; 0.80µm, 1.90, 0.0007].

| Aerosol model | | | |
|---|---|---|---|
| Size distribution | Bimodal lognormal distribution | | |
| | $r_{fine}$ = 0.12 µm | $\sigma_{fine}$ = 1.42 | $N_{fine}$ = 0.9996 |
| | $r_{coarse}$ = 0.62 µm | $\sigma_{coarse}$ = 2.23 | $N_{coarse}$ = 0.0004 |
| Refractive index | 1.51 – 0.029i | | |
| Wavelength | 0.55 µm* | 0.64 µm | 0.81 µm | 1.64 µm |
| SSA | 0.852 | 0.839 | 0.804 | 0.643 |
| g | 0.649 | 0.612 | 0.538 | 0.468 |
| **Cloud model** | | | |
| Size distribution | Gamma law | | |
| | $r_{eff}$ from 4 to 60 µm | $v_{eff}$ = 0.06 | |

**Table 1:** Aerosol and cloud properties used to compute the radiances LUT of the SEVIRI
retrieval. (* Note that 0.55µm does not correspond to a SEVIRI channel.)

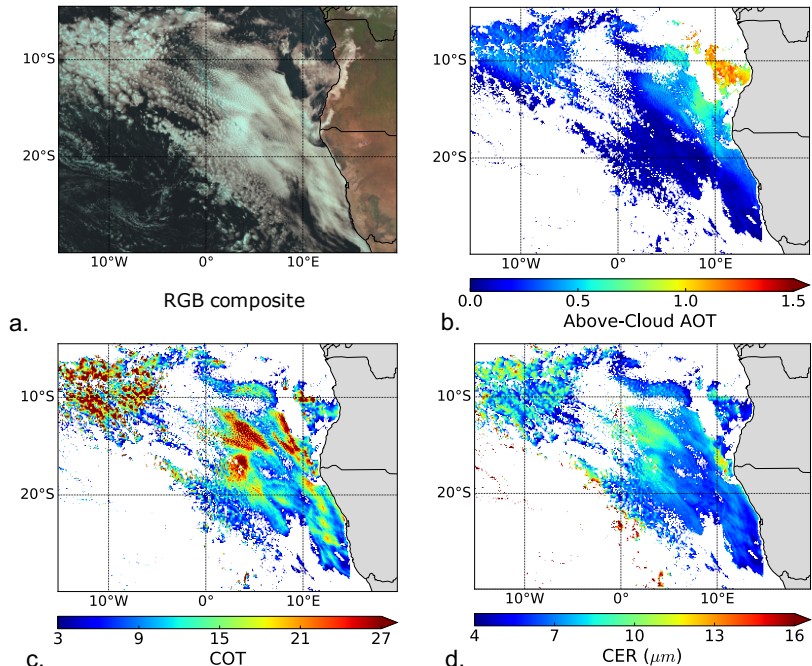

a. RGB composite

b. Above-Cloud AOT

c. COT

d. CER (μm)

**Figure 5:** RGB composite (a), Above cloud AOT at 0.55 μm (b) and cloud properties (c and d) retrieved from SEVIRI measurements on the 28 August 2017 at 10:12 UTC over the SEAO.

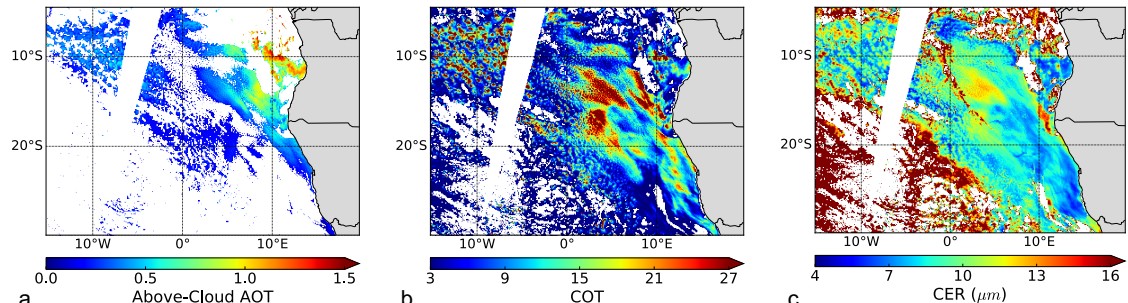

a. Above-Cloud AOT

b. COT

c. CER (μm)

**Figure 6:** Above cloud AOT at 0.55 μm (a) and cloud properties (b and c) retrieved from MODIS-Terra with the MOD06ACAERO algorithm (Meyer et al., 2015) on the 28 August 2017.

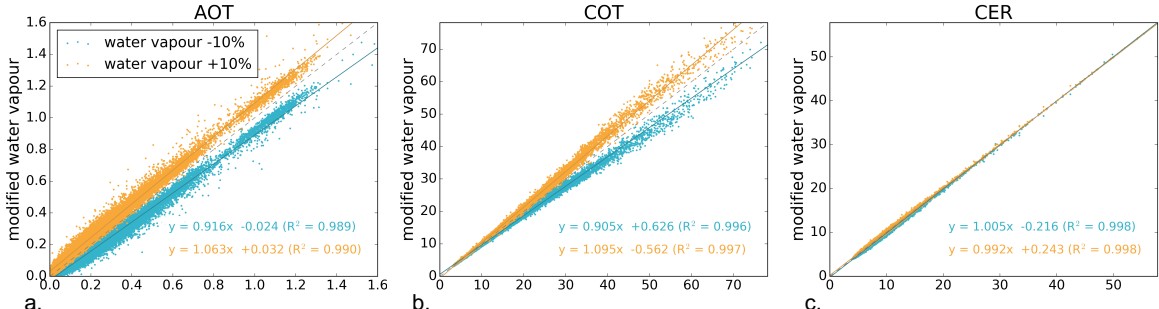

**Figure 7:** Uncertainty in the retrieved above-cloud AOT (a), COT (b) and CER(c) due to an error of +10% in orange and -10% in blue on the specific humidity profile compare to the original forecast for 28 August 2017 at 10:12 UTC.


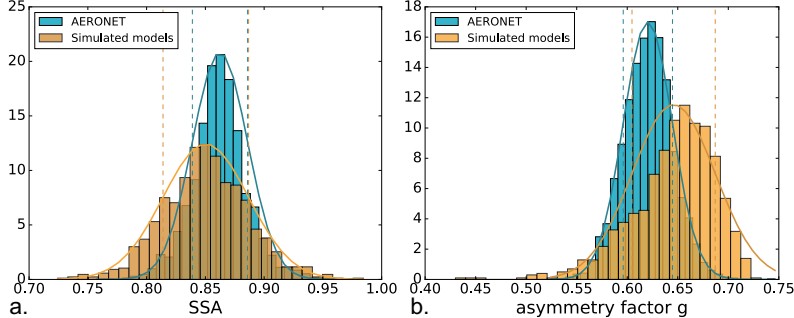


**Figure 8:** Histograms of the SSA (a) and asymmetry factor g (b) at 0.55 µm simulated from a range of size distribution and refractive index (orange) and retrieved by AERONET (blue) over the Southern Africa. Dashed lines represent the mean +/- the standard deviation.


| Model | SSA | g | $r_{fine}$ | $\sigma_{fine}$ | refr. index |
|---|---|---|---|---|---|
| CLARIFY | 0.852 | 0.649 | 0.12 | 1.42 | 1.51-0.029i |
| $SSA_{CLARIFY}-\sigma_{SSA}$ | 0.812 | 0.648 | 0.12 | 1.42 | 1.51-0.037i |
| $SSA_{CLARIFY}+\sigma_{SSA}$ | 0.891 | 0.649 | 0.12 | 1.42 | 1.52-0.021i |
| $g_{CLARIFY}-\sigma_g$ | 0.852 | 0.603 | 0.12 | 1.30 | 1.53-0.027i |
| $g_{CLARIFY}+\sigma_g$ | 0.851 | 0.686 | 0.12 | 1.51 | 1.50-0.030i |
| $SSA_{CLARIFY}-\sigma_{SSA}, g_{CLARIFY}-\sigma_g$ | 0.813 | 0.604 | 0.11 | 1.37 | 1.52-0.034i |
| $SSA_{CLARIFY}+\sigma_{SSA}, g_{CLARIFY}+\sigma_g$ | 0.886 | 0.687 | 0.13 | 1.50 | 1.49-0.022i |
| $SSA_{CLARIFY}-\sigma_{SSA}, g_{CLARIFY}+\sigma_g$ | 0.814 | 0.684 | 0.12 | 1.51 | 1.50-0.041i |
| $SSA_{CLARIFY}+\sigma_{SSA}, g_{CLARIFY}-\sigma_g$ | 0.884 | 0.602 | 0.11 | 1.36 | 1.49-0.017i |


**Table 2:** Aerosol properties used to test the sensitivity of the SEVIRI to the aerosol model. SSA and g are given at 0.55 µm.


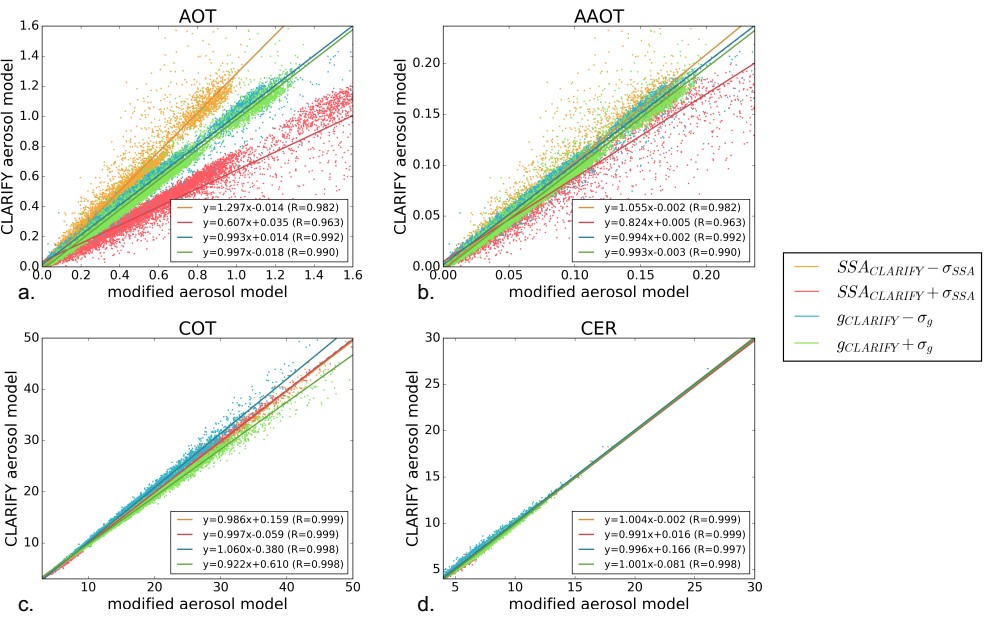


**Figure 9:** Impact of the assumption on the SSA and the asymmetry factor g on the retrieved aerosol and cloud properties. AOT, AAOT, COT and CER obtained for 28 August 2017 at

10:12 UTC with the CLARIFY-2017 model are plotted against the properties retrieved with
the modified aerosol models.

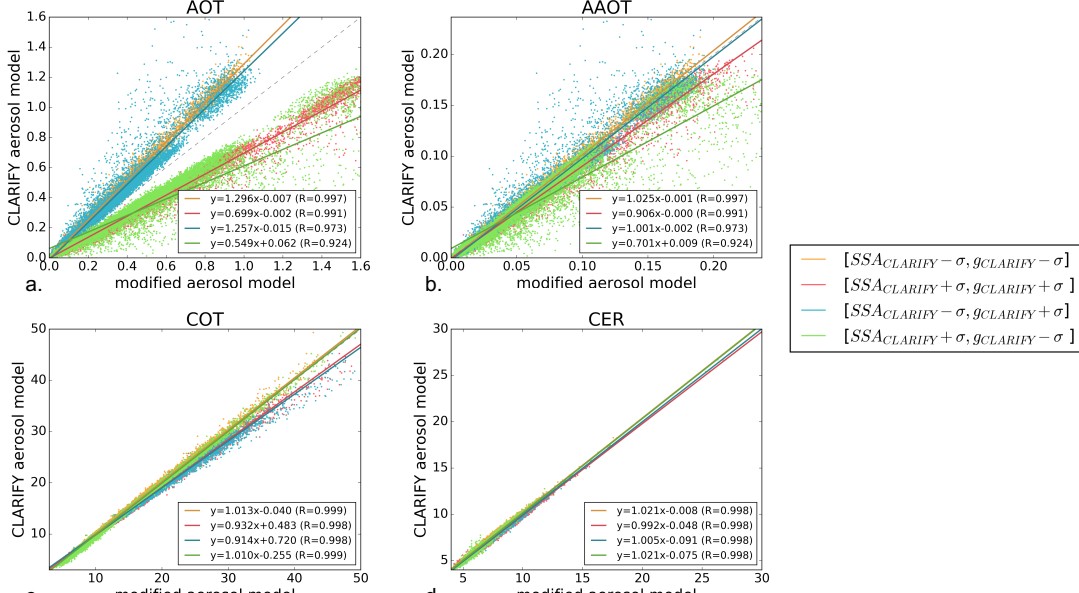

**Figure 10:** Similar to Figure 9 for the combined impact of g and the SSA.

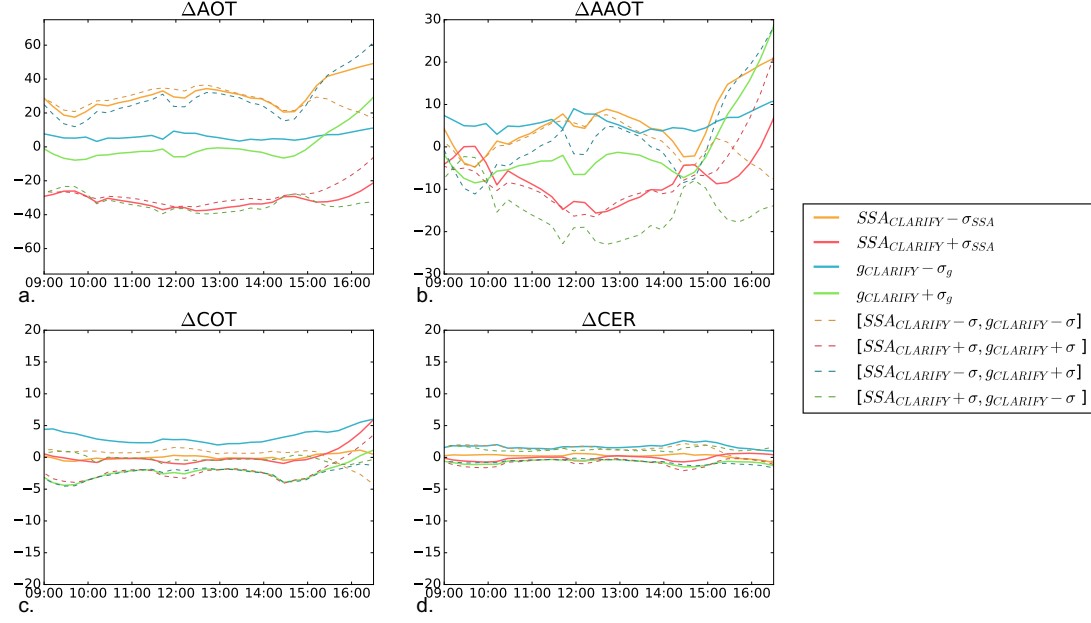

**Figure 11:** Time series (UTC) of the difference Δ (in %) of the above-cloud AOT (a), AAOT
(b), COT (c), CER (d) retrieved with the CLARIFY model and the modified aerosol models
for the 28 August 2017.

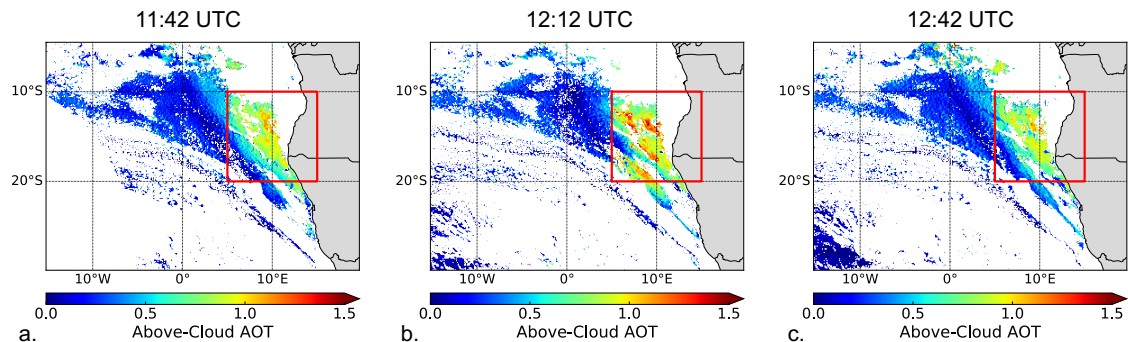

**Figure 12:** Above-cloud AOT retrieved the 05 September 2017 at 11:42, 12:12 and 12:42
UTC. The red square represents the area over which the SEVIRI products have been averaged.

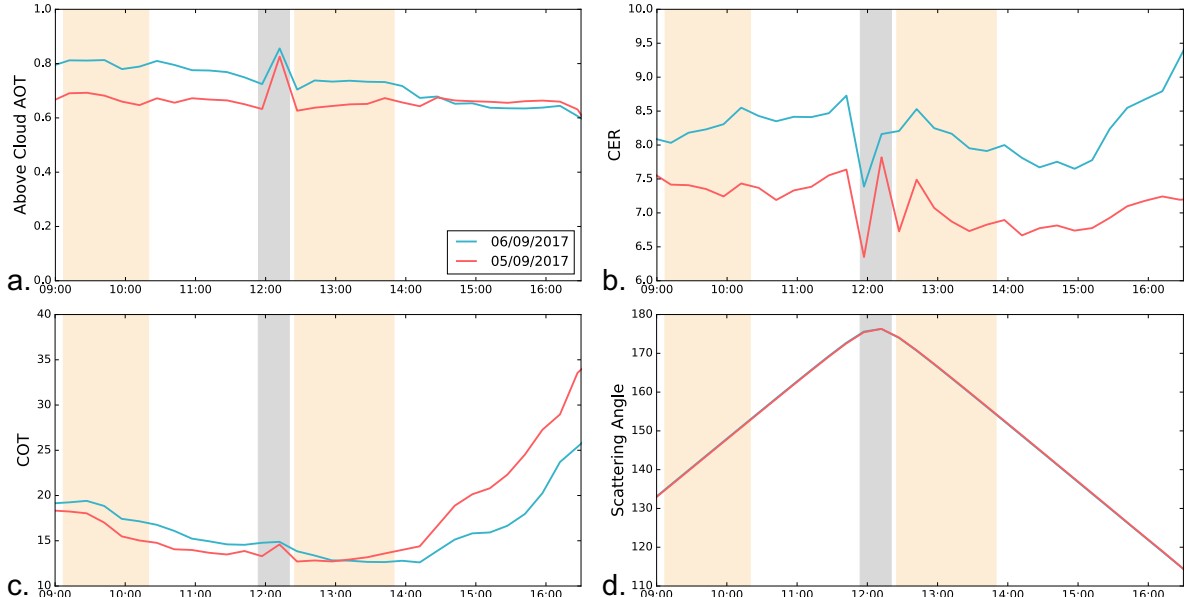

**Figure 13:** Time series (UTC) of the above-cloud AOT (a), COT (b), CER(c) and scattering
angle(d) averaged between 20˚S and 10˚S, and 5˚E and 15˚E for the 5th and 6th September
2017. The grey area represents scattering angles larger than 175˚ and the orange areas show
the typical overpass times of MODIS Aqua and Terra over the region.

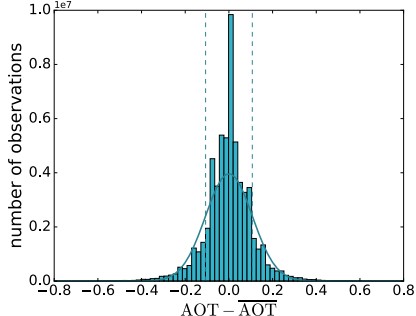

**Figure 14:** Histogram of the difference between AOT retrieved at t=0 and the running mean
calculated between t-15 and t+15 minutes from 01 September to 12 September 2017.
Observations within the glory region have been removed. Dashed lines represent the mean +/-
the standard deviation.