# Peer review of "Observation of absorbing aerosols above clouds over the South-East Atlantic Ocean from the geostationary satellite SEVIRI Part 1: Method description and sensitivity"

_Atmospheric Chemistry and Physics, 2018_

## Referee Comment (RC1) · Ian Chang (Referee) · 8 Feb 2019

Dear Editor:

This manuscript outlines a technique to simultaneously retrieve above-cloud aerosol optical properties and underlying cloud properties from Meteosat Second Generation (MSG) Spinning Enhanced Visible and Infrared Imager (SEVIRI) over the southeast Atlantic. This work demonstrates a cogent pathway for estimating the aerosol direct radiative effects in the southeast Atlantic by using high temporal resolution data to

synchronously evaluate diurnal cycles of aerosol and cloud properties. Overall, this paper is concisely and coherently written with minor technical issues. Thus, I support the publication of this manuscript in the Special issue of Atmos. Chem. Phys. upon addressing the comments and suggestions.

Best regards,

-Ian Chang

General Comments:

The criteria for rejecting aggregated retrievals using standard deviations of AOT and in-homogeneity parameters of CER to remove high AOT uncertainty grids are performed to ensure that the accepted retrievals are reliable. However, such filtering would discard some reliable pixels. I suggest the authors discuss the number of cloudy pixels that are removed using this filtering technique since removing an excessive number of cloudy pixels may have a significant impact on estimating the above-cloud aerosol direct radiative effects. Also, have the authors tested the filtering at finer grid resolutions in order to retain a higher number of reliable retrievals? Despite the remarks, I anticipate that Part 2 of this manuscript will elaborate on these points.

A major objective of this paper addresses the sensitivity of retrievals due to aerosol model assumptions. This analysis is presented using a case study from 28 August 2017 at 1012 UTC. Since this paper aims to demonstrate the validity of simultaneous above-cloud AOT and underlying COT retrievals throughout the day, a sensitivity analysis should be presented at different times of the day instead of only at a particular time of the day. Hence, the authors should present these details during other hours of the day (unless the uncertainty variations are negligible throughout the day) if there are sufficient time and space to consolidate this information. Alternatively, the authors need to explicitly indicate that this uncertainty estimate is limited to a case study and discuss the anticipated uncertainties during other times of the day. The abstract should state the ranges of modified parameters that are used to conduct the sensitivity analysis and

mention the time period that the uncertainties represent.

Specific Comments:

Page 7 Lines 267-269: Is the negligible retrieval sensitivity associated with aerosol/cloud altitude assumptions quantified or is the negligibility a mere presumption? Both Jethva et al. (2013) and Meyer et al. (2015) have quantified retrieval uncertainties associated with aerosol top height assumptions.

Page 10 Lines 398-399: The above-cloud AOT retrievals are stable within two times the standard deviation of the retrievals but not necessarily stable within one standard deviation. Thus, it is only more stable relative to one standard deviation. I suggest the authors justify the validity of defining the stability with respect to two standard deviations.

Technical Corrections:

Page 2 Line 84: "polar orbiting" => "polar-orbiting"

Page 3 Line 89: "from satellite platforms than currently available" => from geostationary satellite platforms instead of polar-orbiting satellite platforms that have coarser temporal resolutions.

Page 4 Line 160: "MODIS, and hence" => "MODIS. Hence, SEVIRI is significantly"

Page 5 Line 174: Remove "and" and "one"

Page 5 Line 212: "of hydration" seems redundant in this sentence.

Page 6 Line 220: "are" => "include"

Page 6 Line 255: "are close" => "are close to each other"

Page 7 Line 268: "due of" => "due to"

Page 7 Line 286: "around" => "approximately"

Page 7 Line 286: "observations" is vague in the context of this sentence. I suggest "pixels" as a more suitable word.

Page 8 Line 297: "on the 28" => "on 28"

Page 8 Line 334: "10% indicating" => "10%, indicating"

Page 9 Line 380: It would be helpful to mention that the uncertainty of each component is computed from the averaged absolute values between the positive and the negative biases of the modified parameter.

Page 10 Line 420: "from" => "for"

Page 11 Line 431: "polar orbiting" => "polar-orbiting"

Page 11 Line 435: "in" => "of"

Page 11 Line 439: "the" => "their"

Page 11 Line 441: "contribution" is a bit vague. I suggest replacing this term with "absorption" or a more definitive term.

Page 12 Line 485: "above cloud" => "above-cloud"

Page 18 Line 729: "Cloud optical thicknesses (COT) and aerosol optical thicknesses (AOT)" => "COTs and AOTs"

Page 18 Line 733: "COT and CER" => "COTs and CERs"

Page 18 Line 734: ""absorbing aerosols above" => "overlying absorbing aerosols"

Page 20 Line 748: "ones" => "lines"

Page 21 Line 758: "composite" => "composite for"

Page 22 Line 779: Remove "the"

Page 24 Line 798: "ones" => "areas"

---

## Referee Comment (RC2) · Anonymous Referee #2 · 11 Feb 2019

General Comments:

The paper describes what looks to be a promising method to simultaneously retrieve above cloud aerosol optical depth with cloud optical depth and effective radius from the Spinning Enhanced Visible and InfraRed Imager (SEVIRI). While the technique used is not tremendously novel, the application to geostationary data appears so, and the recognition of the impact of varying water vapour in particular on the measured satellite signal and hence the retrieved quantities shows good insight.

[Figure]

My own feeling is that the paper is a little 'thin' and actually would have benefited from including some of the material that I anticipate will be in the companion manuscript. Moreover, even if some of these comparisons are included here, given the title I think the paper has to encompass or at least discuss the full range of likely sensitivities that could be present in order to either show more generalised utility or to identify when the method will work optimally.

If this is done I see no reason why the work should not be published.

Specific remarks:

Some aspects of the methodology are not clear. I assume that in working out the aerosol model parameters you first fit the size distribution, then iteratively adjust the refractive indices until you fit the EXSCALABAR measurements of SSA, assuming that the biomass aerosols are spherical. Is this correct? If so I think you must: (a) provide some error bars on the size distribution and SSA observations in figure 4. These could then perhaps be used to give a realistic range in the size distribution parameters and the complex refractive index that you have selected. At the moment the reader has no feel whether it is sensible to try to match the EXSCALABAR data as well as you have. (b) justify the assumption of Mie scattering

Does EXSCALABAR extend further than 0.65 microns? This would give more confidence in the final aerosol model both in terms of the size distribution and complex refractive index at the longer SEVIRI channel wavelengths. The assumption of a fixed refractive index with wavelength seems quite large.

You seem to assume a fixed aerosol and cloud layer height. Is this realistic and what impact does it have if the 'real' heights are different (i.e. did you actually investigate the impact of varying these heights – you imply it is negligible)?

It is good that you have investigated the impact of variations in humidity on the retrievals via your correction process but you are limited to the baseline set of atmospheres

contained in the case study you have selected. Are the retrieval errors likely to be of the same order of magnitude if these conditions change? Or how sensitive are you to both the total amount and vertical distribution of water vapour? What about uncertainty in the cloud top height (line 180)? I believe it is quite challenging to (a) detect and (b) accurately locate low cloud over ocean using thermal IR radiances.

Similarly, are you sure that you have considered a wide enough variation in aerosol model parameters? You don't really justify the choices that are made for the perturbations applied. Lines 364 and 365 imply that there should be a variation in the aerosol properties in the study region but you don't tie these to the perturbations you have implemented.

AAOT rather appears from nowhere at line 350. I think it would benefit from at least a small introduction. Before this, all the focus has been on AOT. Line 373 implies that changing the imaginary part of the refractive index results in a very large perturbation to the AOT retrieval (where does the 39 % actually come from – not obvious from the scatter plots which have points that look like there is a higher difference). You imply that the impact is much smaller on the AAOT but do not really clearly explain why. I think I have worked it out but it is not immediately apparent from the text so I suggest a little rewrite here.

Are you sure that your uncertainty terms in equation 4 are independent? I would think not given how (I think) the size distribution and refractive indices have been derived. Moreover, even if they are independent, this is only the uncertainty due to the aerosol model. Uncertainty in the water vapour correction (and cloud top height) will also inflate the uncertainty in the final retrievals. Are these combined anywhere?

In line 382 you state the aerosol model uncertainty as 31 %. It's not immediately obvious how this is consistent with your earlier statement that the uncertainty from the imaginary part of the refractive index can reach 39 % so how do you arrive at this number (could be due to absolute values but it would be nice to be clear)?

I find the evaluation of the retrievals a little lacking. The comparisons to AERONET and MODIS in section 3(a) are very qualitative. It seems obvious to at least include the equivalent MODIS retrievals in figure 12 simply to give some idea of the quantitative consistency between these and the SEVIRI estimates even if it is not clear which, if either, estimate is correct. This should still leave plenty of scope to enlarge on these comparisons in the planned companion paper.

I think the linear trends in Figure 12 add nothing. I'd much prefer to see the individual standard deviations and perhaps even the estimated uncertainty (which are not the same).

Technical Corrections:

At some point early in the manuscript please identify the wavelength(s?) of the COT and AOT retrievals.

Line 48: You've been talking about effect but here you mention forcing. They are not the same. 'of up to'

Line 59: Here I think you are talking about the aerosol indirect effect. It would good to say this explicitly for consistency with the next sentence.

Line 70: Not sure why 'Aerosols Above Clouds' is capitalised.

Line 86: '. . .cloud cover over the SEAO has an. . ..'

Line 124: I appreciate the terms may have been defined elsewhere but I think it would be good to explicitly give the definition here.

Line 129: follows

Line 132: actually from figure 1 there does seem to be some dependence on COT.

Line 143: 'increases the SWIR'. Actually you do not explicitly define NIR and SWIR in terms of wavelength range. This would be helpful. Or lose the terms entirely and just

use the wavelengths.

Line 166: please explain 'two-way transmittance' – from where to where? Why is the two-way transmittance important?

Line 228-233: Not really enough detail on 'weighting'. Someone would struggle to replicate what you have done from this info alone.

Line 255: For the uninitiated it might be useful to say where SAFARI was.

Line 309: 'typically observed in this region' – as shown by who exactly?

Line 322: Can you provide a reference for this statement please.

Line 473-474: This isn't immediately obvious to me. Can you clarify? Obviously you could use a different aerosol model in the LUT but this wouldn't be 'easy'.

Figure 5(b): You have lost the latitude labels

Figure 7-11: y-axes labels. Suggest adding 1-1 lines.

Figure 12: Add time basis (e.g. UTC).

---

## Author Comment (AC1) · 10 May 2019

**Response to reviews for acp-2018-13333**

We are thankful to Dr. Ian Chang and the anonymous referee for evaluating this manuscript and helping us to improve its quality with their feedback. Please, find below our responses to reviewer comments, with our responses in black text.

Ian Chang (Referee)

Dear Editor:

This manuscript outlines a technique to simultaneously retrieve above-cloud aerosol optical properties and underlying cloud properties from Meteosat Second Generation (MSG) Spinning Enhanced Visible and Infrared Imager (SEVIRI) over the southeast Atlantic. This work demonstrates a cogent pathway for estimating the aerosol direct radiative effects in the southeast Atlantic by using high temporal resolution data to synchronously evaluate diurnal cycles of aerosol and cloud properties. Overall, this paper is concisely and coherently written with minor technical issues. Thus, I support the publication of this manuscript in the Special issue of Atmos. Chem. Phys. upon addressing the comments and suggestions.

Best regards,
-Ian Chang

**General Comments:**

The criteria for rejecting aggregated retrievals using standard deviations of AOT and inhomogeneity parameters of CER to remove high AOT uncertainty grids are performed to ensure that the accepted retrievals are reliable. However, such filtering would discard some reliable pixels. I suggest the authors discuss the number of cloudy pixels that are removed using this filtering technique since removing an excessive number of cloudy pixels may have a significant impact on estimating the above-cloud aerosol direct radiative effects. Also, have the authors tested the filtering at finer grid resolutions in order to retain a higher number of reliable retrievals? Despite the remarks, I anticipate that Part 2 of this manuscript will elaborate on these points.

The filters described in the manuscript have been implemented in order to ensure that the measurements have been performed in optimum conditions for the retrieval. The LUT has been computed with a 1D radiative transfer code. At cloud edges and for inhomogeneous clouds, the independent pixel approximation is not valid and the plane-parallel bias is not negligible. Therefore, the retrieval of the aerosol and cloud properties becomes unstable, and using those data could lead to significant errors in the estimation of the above-cloud aerosol direct radiative effect. Removing those observations allows us to improve the quality of the final products. For the case study described in the manuscript (28/08/2017 at 10:12 UTC), those removed observations represent 23.7% of the pixels. Figure 1 in this document shows the AOT retrieved above clouds together with the filtered pixels in magenta. We note that while removing these pixels is not ideal when comparing to e.g. GCM model studies, by clearly stating our assumptions, GCM studies can perform a similar screening procedure.

[Figure]

**Figure 1:** Above cloud AOT at 550 nm retrieved from SEVIRI measurements on the 28 August 2017 at 10:12 UTC over the SEAO. Pixels in magenta correspond to pixels removed with the cloud edge and cloud heterogeneity filters.

A major objective of this paper addresses the sensitivity of retrievals due to aerosol model assumptions. This analysis is presented using a case study from 28 August 2017 at 1012 UTC. Since this paper aims to demonstrate the validity of simultaneous above-cloud AOT and underlying COT retrievals throughout the day, a sensitivity analysis should be presented at different times of the day instead of only at a particular time of the day. Hence, the authors should present these details during other hours of the day (unless the uncertainty variations are negligible throughout the day) if there are sufficient time and space to consolidate this information. Alternatively, the authors need to explicitly indicate that this uncertainty estimate is limited to a case study and discuss the anticipated uncertainties during other times of the day. The abstract should state the ranges of modified parameters that are used to conduct the sensitivity analysis and mention the time period that the uncertainties represent.

The entire Section 3c has been modified. The evolution of the sensitivity of the retrieval to the aerosol model assumptions during the day has been analysed. The uncertainty on the cloud properties remains small all day long (lower than 5.6% for the COT and 2.6% on the CER), with the sensitivity of the COT being slightly smaller in the middle of the day. We also show that the uncertainty on the retrieved AOT and AAOT is smallest during the 09:00-15:00 UTC time period. The following analysis has been added to Section 3c:

*The variation of the solar zenith angle, and therefore, of the satellite observation geometry during the day can impact the sensitivity of the retrieval to the aerosol assumptions. Therefore, the 15-minute SEVIRI observations for the 28 August have been processed using the eight aerosol models described above and compared to the aerosol and cloud properties retrieved with the CLARIFY aerosol model. The difference $\Delta x_i$ of a product x is defined as:*

$$\Delta x_i = (x_{CLARIFY} - x_i)/x_i \times 100\%$$

*where $x_{CLARIFY}$ and $x_i$ is the mean product x retrieved over the SEVIRI slot with the aerosol CLARIFY model and the modified model i, respectively. Figure 11 shows the time series of $\Delta AOT$ (a), $\Delta AAOT$ (b), $\Delta COT$ (c) and $\Delta CER$ (d) obtained with the modified aerosol models. The sensitivity of the retrieved cloud properties to the aerosol model assumptions remains small (lower than 5.6% for the COT and 2.6% for the CER) and dominated by the sensitivity*

*to g. Apart from a small decrease of ΔCOT at midday when g is overestimated (solid blue line) and an increase of ΔCOT in late afternoon when the SSA is underestimated (solid red line), no significant trend is observed on the cloud property sensitivities. As observed previously, the uncertainty on the AOT is led by the SSA assumption, with the AOT being overestimated (respectively underestimated) when the assumed SSA is overestimated (respectively underestimated). Until 15:00, ΔAOT stays within +/-40%, with the sensitivity to the SSA being slightly larger at midday. Then it increases up to 60% when the SSA is overestimated and g is underestimated (dashed blue line). Similar trends are observed on ΔAAOT, with generally lower values than ΔAOT. An increase of the uncertainty is observed on the AAOT after 15:00, that reaches up to 27% at 16:30. Before 15:00, there is a larger AAOT sensitivity to the SSA around midday (+8.9%/-15.2%), but there is no evident evolution of the sensitivity to g with time. The case that lead to the largest biases on the AAOT is when the SSA is underestimated and g overestimated (dashed green lines), with an underestimation of up to 23%. However, it should be noted that 0% of the AERONET observations used in Figure 8 are associated with an SSA lower than $SSA_{CLARIFY}$-$\sigma_{SSA}$ and a g larger than $g_{CLARIFY}$-$\sigma_g$. Otherwise, the sensitivity of the AAOT to the aerosol property assumptions stays between -16.6 and +9% before 15:00.*

*In conclusion, the retrieved AOT is less sensitive to the aerosol property assumption before 15:00, with an uncertainty of 40%. This uncertainty is dominated by the sensitivity of the retrieval to the SSA. An overestimation (respectively underestimation) of the AOT is expected when the observed aerosols are more (respectively less) absorbing than the aerosol model assumed for the retrieval. A better accuracy is obtained on the retrieved AAOT, with an uncertainty generally lower than 17 % before 15:00. The sensitivity of the cloud properties to the aerosol model assumption remain small all day long, with an uncertainty of 5.6% on the COT and 2.6% on the CER.*

In the conclusion, the following text has been added:
*Retrievals have been performed considering aerosol models with modified SSA and asymmetry factor g. It has been shown that the sensitivity of the retrieved cloud properties to the aerosol model assumption is small with errors lower than 5.6% on the COT and 2.6% on the CER. As expected the impact of the assumed aerosol properties is much larger on the above cloud AOT, with an uncertainty estimated at 40% before 15:00 UTC.*

Finally, the comments about the sensitivity analysis in the abstract have been modified:
*Between 09:00-15:00 UTC, an uncertainty of 40% is estimated on the above-cloud AOT, which is dominated by the sensitivity of the retrieval to the single scattering albedo. The absorption AOT is less sensitive to the aerosol assumptions with an uncertainty generally lower than 17% between 09:00-15:00 UTC. Outside of that time range, as the scattering angle decreases, the sensitivity of the AOT and the absorption AOT to the aerosol model increases. The retrieved cloud properties are only weakly sensitive to the aerosol model assumptions throughout the day, with biases lower than 6% on the COT and 3% on the CER.*

**Specific Comments:**

Page 7 Lines 267-269: Is the negligible retrieval sensitivity associated with aerosol/cloud altitude assumptions quantified or is the negligibility a mere presumption? Both Jethva et al.

(2013) and Meyer et al. (2015) have quantified retrieval uncertainties associated with aerosol top height assumptions.

The Rayleigh scattering is expected to be small at the wavelengths used for the retrieval. A test has been made using new LUT, assuming a cloud top height at 3 km and an aerosol layer located between 4 and 5 km. An impact of 2.3% has been observed on the AOT and lower than 0.3% on the cloud properties. The following sentence has been added to section 2d:
*We have evaluated the error due to the fixed aerosol and cloud altitudes to be lower than 2.5% on the AOT and 0.3% on the cloud properties.*

Page 10 Lines 398-399: The above-cloud AOT retrievals are stable within two times the standard deviation of the retrievals but not necessarily stable within one standard deviation. Thus, it is only more stable relative to one standard deviation. I suggest the authors justify the validity of defining the stability with respect to two standard deviations.

The obvious benefit of geostationary SEVIRI retrievals over polar-orbiting satellite retrievals are that they are available every 15minutes. It is therefore relevant to examine whether the retrievals made at time t=0 and at time t+15mins are similar; if they were not, then this would suggest that the retrieval algorithm is not stable. Note that this assumes that the scene is changing relatively slowly; cloud and aerosol optical depths should not vary between time t=0 and time t+15minutes. Figure 13 indicates that outside of the glory regions, the retrieval algorithm does indeed appear to be stable; there is little variation from one time step to the next. We included the 2sd measure as a rough metric, but on reflection it adds little to our quantification of stability as we only applied this metric to the above cloud AOT. This is because there can be longer timescale trends in cloud and aerosol (indeed there is a shallow slope in the above cloud AOT for 06/09/2017 of around 0.02 to 0.03/hour), and these statistics would differ depending on the area chosen and the prevailing meteorological conditions. Therefore, we chose to remove this statistical analysis as it is too simplistic.

**Technical Corrections:**

Page 2 Line 84: "polar orbiting" => "polar-orbiting"

The correction has been applied.

Page 3 Line 89: "from satellite platforms than currently available" => from geostationary satellite platforms instead of polar-orbiting satellite platforms that have coarser temporal resolutions.

The sentence has been modified to:
*Therefore, the study of the SEAO cloud and above-cloud aerosol optical properties would benefit from the high temporal resolution observations provided by geostationary satellite platforms.*

Page 4 Line 160: "MODIS, and hence" => "MODIS. Hence, SEVIRI is significantly"

Page 5 Line 174: Remove "and" and "one"

Page 5 Line 212: "of hydration" seems redundant in this sentence.

Page 6 Line 220: "are" => "include"

Page 6 Line 255: "are close" => "are close to each other"

Page 7 Line 268: "due of" => "due to"

Page 7 Line 286: "around" => "approximately"

Page 7 Line 286: "observations" is vague in the context of this sentence. I suggest "pixels" as a more suitable word.

Page 8 Line 297: "on the 28" => "on 28"

Page 8 Line 334: "10% indicating" => "10%, indicating"

Page 9 Line 380: It would be helpful to mention that the uncertainty of each component is computed from the averaged absolute values between the positive and the negative biases of the modified parameter.

Page 10 Line 420: "from" => "for"

Page 11 Line 431: "polar orbiting" => "polar-orbiting"

Page 11 Line 435: "in" => "of"

Page 11 Line 439: "the" => "their"

Page 11 Line 441: "contribution" is a bit vague. I suggest replacing this term with "absorption" or a more definitive term.

Page 12 Line 485: "above cloud" => "above-cloud"

Page 18 Line 729: "Cloud optical thicknesses (COT) and aerosol optical thicknesses (AOT)" => "COTs and AOTs"

Page 18 Line 733: "COT and CER" => "COTs and CERs"

Page 18 Line 734: "absorbing aerosols above" => "overlying absorbing aerosols"

Page 20 Line 748: "ones" => "lines"

Page 21 Line 758: "composite" => "composite for"

Page 22 Line 779: Remove "the"

Page 24 Line 798: "ones" => "areas"

The above corrections have been applied.

Anonymous Referee #2

**General Comments:**

The paper describes what looks to be a promising method to simultaneously retrieve above cloud aerosol optical depth with cloud optical depth and effective radius from the Spinning Enhanced Visible and InfraRed Imager (SEVIRI). While the technique used is not tremendously novel, the application to geostationary data appears so, and the recognition of the impact of varying water vapour in particular on the measured satellite signal and hence the retrieved quantities shows good insight.

My own feeling is that the paper is a little 'thin' and actually would have benefited from including some of the material that I anticipate will be in the companion manuscript. Moreover, even if some of these comparisons are included here, given the title I think the paper has to encompass or at least discuss the full range of likely sensitivities that could be present in order to either show more generalised utility or to identify when the method will work optimally. If this is done I see no reason why the work should not be published.

**Specific remarks:**

Some aspects of the methodology are not clear. I assume that in working out the aerosol model parameters you first fit the size distribution, then iteratively adjust the refractive indices until you fit the EXSCALABAR measurements of SSA, assuming that the biomass aerosols are spherical. Is this correct? If so I think you must: (a) provide some error bars on the size distribution and SSA observations in figure 4. These could then perhaps be used to give a realistic range in the size distribution parameters and the complex refractive index that you have selected. At the moment the reader has no feel whether it is sensible to try to match the EXSCALABAR data as well as you have. (b) justify the assumption of Mie scattering.

(a) Errors bars are now provided in Figure 4 in the manuscript and the following explanations have been added to Section 2c:

*The uncertainty in SSA calculations are related to the corresponding uncertainties in the extinction and absorption coefficients measured by EXSCALABAR. This error analysis has been performed previously and the reader is directed to Davies et al. (2019). Briefly, the measured extinction has an accuracy of ~2%, and we use a 2% extinction uncertainty in the analysis here. The errors in absorption measurements using photoacoustic spectroscopy depend on uncertainties in the ozone calibration, microphone pressure dependence and the background response from laser scattering/absorption on the windows of the photoacoustic cell. We have shown in recent publications that our calibration uncertainties are ~5% (Cotterell et al. 2019; Davies et al. 2018), and the uncertainty in the pressure-dependent microphone response is 1.2% (Davies et al. 2019). The background response from laser-window interactions is ranging from 0.27 – 0.54 $Mm^{-1}$. Thus, the total absorption uncertainty, propagating all the above uncertainties, is absorption-dependent and ranges from 29.0 – 55.0 % (dependent on PAS measurement wavelength) at 1 $Mm^{-1}$ and 8.1 % at 100 $Mm^{-1}$ (independent of PAS measurement wavelength). We propagated these total measurement uncertainties for both extinction and absorption measurements to derive the standard deviation σ in our calculated SSA values. We find that the mean SSA uncertainties are 0.013 and 0.018 at the measurement wavelengths of 405 and 658 nm respectively.*

*Three sources of errors have been taken into account on the PCASP measurements: the error on the bin concentration is calculated according to Poisson counting statistics, the sample flow rate error is assumed to be 10% and a bin edge calibration error of half a bin has been considered.*

*The aerosol model is selected by iteratively adjusting the refractive index and fitting the PCASP measurements (Fig. 4a) until the aerosol model matches the SSA from EXSCALABAR (Fig. 4b).*

*The uncertainties on the aerosol properties have been estimated using the errors on the PCASP and EXSCALABAR measurements. The uncertainty on the imaginary part of the refractive index is 0.02 for the real part and 0.004 for the imaginary part. For the size distribution, the uncertainty is 0.016μm, 0.09 and 0.00045 for radius, the standard deviation and the number fraction of the fine mode respectively.*

(b) Martins et al. have observed that smoke particles from biomass burning could be considered spherical one hour after being emitted, which justify the use of the Mie theory. The following sentence has been added to Section 2c:

*The aerosol optical properties are calculated using the Mie theory, as the spherical approximation is expected to be valid for biomass burning particles from one hour after being released in the atmosphere (Martins et al., 1998).*

Martins, J. V., Hobbs, P. V., Weiss, R. E., and Artaxo, P., Sphericity and morphology of smoke particles from biomass burning in Brazil, Journal of Geophysical Research, 103( D24), 32051–32057, doi:10.1029/98JD01153, 1998.

Does EXSCALABAR extend further than 0.65 microns? This would give more confidence in the final aerosol model both in terms of the size distribution and complex refractive index at the longer SEVIRI channel wavelengths. The assumption of a fixed refractive index with wavelength seems quite large.

EXSCALABAR does not extend further than 0.65μm. However, shortwave irradiance spectra from 0.3 to 1.7 μm were measured during the campaign with the SHIMS (Spectral Hemispheric Irradiance Measurements) instrument. The radiative closure using the CLARIFY aerosol model is being studied and a paper is currently in preparation. The assumption of a fixed refractive index with wavelength has been motivated by the relatively small contribution of aerosols from the coarse mode and therefore, by the small impact of aerosols on the satellite signal measured at 1.6μm. For the algorithm developed by Meyer et al. (2015), which uses a similar method and spectral bands at longer wavelengths, the aerosol refractive index is also assumed to be spectrally invariant.

Meyer, K., Platnick, S., and Zhang, Z.: Simultaneously inferring above-cloud absorbing aerosol optical thickness and underlying liquid phase cloud optical and microphysical properties using MODIS, Journal of Geophysical Research: Atmospheres, 120, 5524–5547, https://doi.org/10.1002/2015JD023128, 2015.

*You seem to assume a fixed aerosol and cloud layer height. Is this realistic and what impact does it have if the 'real' heights are different (i.e. did you actually investigate the impact of varying these heights – you imply it is negligible)?*

Next to the coast, where the AOT is usually the largest, the cloud top derived from CALIOP and CATS is usually around 1 km (Rajapakshe et al., 2017). It slightly rises to the west, reaching 1.5/2.0 km at 19W. The satellite observations indicate that the bottom of the aerosol layer is within 2 and 3.5 km and the top is between 3 and 5 km. However, the Rayleigh scattering is expected to have a small contribution to the signal at the wavelength used for the retrieval. The influence of the fixed aerosol and cloud altitudes has been investigated by processing new LUT with a cloud top height at 3 km and an aerosol layer located between 4 and 5 km. The impact of the AOT is estimated to be lower than 2.5% and the impact on the cloud properties is lower than 0.3%. The following sentence has been added to section 2d:
*We have evaluated the error due to the fixed aerosol and cloud altitudes to be lower than 2.5% on the AOT and 0.3% on the cloud properties.*

Rajapakshe, C., Zhang, Z., Yorks, J. E., Yu, H., Tan, Q., Meyer, K., ... & Winker, D. M. Seasonally transported aerosol layers over southeast Atlantic are closer to underlying clouds than previously reported. Geophysical Research Letters, 44(11), 5818-5825, 2017.

*It is good that you have investigated the impact of variations in humidity on the retrievals via your correction process but you are limited to the baseline set of atmospheres contained in the case study you have selected. Are the retrieval errors likely to be of the same order of magnitude if these conditions change? Or how sensitive are you to both the total amount and vertical distribution of water vapour? What about uncertainty in the cloud top height (line 180)? I believe it is quite challenging to (a) detect and (b) accurately locate low cloud over ocean using thermal IR radiances.*

In the companion paper, a section will be dedicated to the validation of the atmospheric correction scheme. The water vapour profiles from the forecast have been compared with the dropsonde measurements from the CLARIFY campaign. Figure 2 in this document shows comparison of the column integrated water vapour. In order to be consistent with the atmospheric correction scheme, the integration of the forecasted water vapour above cloud is done based on the cloud top height retrieved by SEVIRI. Note that the measurements from the CLARIFY dropsondes have not been assimilated in the forecast model. In general, there is a relatively good agreement between the observations and the forecast, especially above clouds. We have also looked at the tephigrams obtained from the forecast and the measurements. An example is shown in Figure 3 of this document. A good consistency is generally obtained for the vertical distribution of the water vapour. On the analysed profiles, we have observed that the cloud top heights retrieved by SEVIRI using the thermal IR radiances are consistent with the altitude of the temperature inversion from the forecast model.

[Figure]

**Figure 2:** Comparison of the above cloud and the full column integrated water vapour from the dropsondes and from the NWP forecast.

[Figure]

**Figure 3:** Tephigram obtained from the dropsonde (dashed lines) and the forecast (solid lines) for the flight C051 of the CLARIFY campaign.

*Similarly, are you sure that you have considered a wide enough variation in aerosol model parameters? You don't really justify the choices that are made for the perturbations applied. Lines 364 and 365 imply that there should be a variation in the aerosol properties in the study region but you don't tie these to the perturbations you have implemented.*

We have modified the analysis of the sensitivity of the retrieval to the aerosol assumption. Instead of analysing the sensitivity to the aerosol size distribution and refractive index separately, we considered a range of SSA and asymmetry factor g that is consistent with observations from AERONET. The choice of the variation in aerosol model parameters and the result of the sensitivity analysis read as follows:

[revised manuscript text omitted]

AAOT rather appears from nowhere at line 350. I think it would benefit from at least a small introduction. Before this, all the focus has been on AOT. Line 373 implies that changing the imaginary part of the refractive index results in a very large perturbation to the AOT retrieval (where does the 39 % actually come from – not obvious from the scatter plots which have points that look like there is a higher difference). You imply that the impact is much smaller on the AAOT but do not really clearly explain why. I think I have worked it out but it is not immediately apparent from the text so I suggest a little rewrite here.

The retrieval is mainly sensitive to the AAOT because it detects the attenuation of the light reflected by the clouds due to the aerosol absorption. Therefore, as the SSA change, the error is expected to primarily affect the scattering AOT. This is why the AOT is more sensitive to the SSA assumption than the AAOT. In section 3c, we added the sentences:

*The retrieval of the above-cloud AOT depends mostly on the aerosol absorption of the light reflected by the cloud. Therefore, it is expected that the retrieved AAOT is less sensitive to the absorbing property of the aerosol than the AOT.*

*This means that a perturbation of the SSA primarily impacts the scattering AOT.*

In Section 2a, the following sentence has been added in order to introduce the AAOT:

*This attenuation is mainly due to the absorption from the aerosol layer, which means that it is primarily correlated to the Absorption AOT (AAOT).*

Are you sure that your uncertainty terms in equation 4 are independent? I would think not given how (I think) the size distribution and refractive indices have been derived. Moreover, even if they are independent, this is only the uncertainty due to the aerosol model. Uncertainty in the water vapour correction (and cloud top height) will also inflate the uncertainty in the final retrievals. Are these combined anywhere?

This part has been removed from the paper. Instead, we have analysed the impact of a perturbation of the SSA and g, both independently and combined. Contrary to the uncertainty on the atmospheric correction, the uncertainties on the aerosol model depends on the assumptions made in the retrieval algorithm. The following sentences have been added at the end of Section 2d:

*It is important to realise that the uncertainties that we quantify here are structural and parametric uncertainties related to assumptions made in the retrieval algorithm. When using a fixed aerosol model, no account is made for natural variability in the aerosol optical parameters and the associated uncertainty; this is dealt with in the uncertainty analysis that follows.*

In line 382 you state the aerosol model uncertainty as 31 %. It's not immediately obvious how this is consistent with your earlier statement that the uncertainty from the imaginary part of the refractive index can reach 39 % so how do you arrive at this number (could be due to absolute values but it would be nice to be clear)?

This part of the manuscript has been removed and we do not used the absolute average difference anymore. The uncertainty on the retrieved property is defined as the difference between the mean property retrieved with the CLARIFY model and with the mean property retrieved with the modified aerosol model.

I find the evaluation of the retrievals a little lacking. The comparisons to AERONET and MODIS in section 3(a) are very qualitative. It seems obvious to at least include the equivalent MODIS retrievals in figure 12 simply to give some idea of the quantitative consistency between these and the SEVIRI estimates even if it is not clear which, if either, estimate is correct. This should still leave plenty of scope to enlarge on these comparisons in the planned companion paper.

In the region of analysis in Section 4, there is a gap between the two MODIS overpasses in the morning of 05 September and in the afternoon of 06 September. In this area, there is a strong gradient of AOT and it is preferable to compare collocated observations.

Instead of the operational MODIS cloud products, the new Figure 6 in the manuscript shows the maps of the equivalent MODIS aerosol and cloud properties from the MOD06ACAERO retrieval. A short description of these results and how they compare to the SEVIRI products has been added to Section 3a:

*As a comparison, Figure 6 shows the equivalent aerosol and cloud properties retrieved from MODIS-Terra with the MOD06ACAERO algorithm (Meyer et al., 2015) for the 10:00 and 11:30 UTC overpasses. The MODIS above-cloud AOT pixels associated with an uncertainty larger than 100% have been removed. A good spatial agreement is observed between the two satellites products. The above-cloud AOT from MODIS is also 1.0 on average close to the coast. On average over the map, the MODIS above-cloud AOT is larger by 0.05 compared to SEVIRI. Considering that MODIS is less sensitive to the atmospheric absorption and that the two algorithms are based on the same principle, the small differences observed between the two above-cloud AOT tend to validate the atmospheric correction applied on the SEVIRI measurements for that case. There is a good consistency between the MODIS and the SEVIRI COT. Finally, the CER retrieved with the MOD06ACAERO algorithm is larger by 2.2 µm compared to the SEVIRI CER. This almost systematic difference is mainly due to differences in the satellite instruments, and especially, the difference in the channels used for the retrieval (Platnick, 2000).*

I think the linear trends in Figure 12 add nothing. I'd much prefer to see the individual standard deviations and perhaps even the estimated uncertainty (which are not the same).

The linear trends have been removed from the figure. In the new version of the manuscript, the stability of the retrieval is also assessed at pixel level by evaluating the variability of the above-cloud AOT between continuous observations. The following analysis has been added at the end of Section 4:

*The performance of the algorithm is further assessed by evaluating the stability of the retrieved above-cloud AOT at pixel level. As noted by Chang and Christopher (2016), in this region over these scales, aerosols are expected to have a limited temporal variability and the variation of the above-cloud AOT is expected to be small between t=0 and t+/-15 minutes. The differences*

*between the AOT retrieved at t=0 and the running mean estimated between t-15 and t+15 minutes have been calculated at pixel level for observations between 09:00-15:00 UTC, removing measurements within the glory backscattering region. Figure 14 shows the histogram of the AOT differences calculated over a 12-day period (01 to 12 September 2017). The differences follow a normal distribution centred around 0.0 with a standard deviation of 0.1. This short-term variability can be attributed to several sources of uncertainties, such as the total amount of water vapour, its vertical distribution, the retrieved cloud top height and the numerical fitting procedure. This analysis indicates that the retrieval of the above-cloud AOT remains relatively stable, with an observed variability of +/- 0.1 between consecutive observations.*

**Technical Corrections:**

At some point early in the manuscript please identify the wavelength(s?) of the COT and AOT retrievals.

Optical thicknesses are given at 0.55μm. This information is now mentioned in Section 3a:

*Throughout this paper, the radiances R refer to the normalized quantity as defined by Herman et al. (2005) and the optical thicknesses (i.e. AOT, COT) are given at 0.55μm.*

Line 48: You've been talking about effect but here you mention forcing. They are not the same. 'of up to'

This sentence has been corrected:

*Positive instantaneous DRE of up to +130W m$^{-2}$ has been observed by satellite instruments over the SEAO (De Graaf et al., 2012; Peers et al., 2015).*

Line 59: Here I think you are talking about the aerosol indirect effect. It would good to say this explicitly for consistency with the next sentence.

This sentence has been modified:

*Biomass burning particles may also have indirect effects through their interactions with cloud droplets, leading to a modification of the microphysics of the cloud, its lifetime and precipitations (Twomey, 1974; Rosenfeld, 2000).*

Line 70: Not sure why 'Aerosols Above Clouds' is capitalised.

Line 86: '. . .cloud cover over the SEAO has an. . ..'

Line 124: I appreciate the terms may have been defined elsewhere but I think it would be good to explicitly give the definition here.

Line 129: follows

These corrections have been applied.

Line 132: actually from figure 1 there does seem to be some dependence on COT.

This sentence has been modified:

*For AOT = 0, the radiance ratio is around 1 and weakly depends on the COT.*

Line 143: 'increases the SWIR'. Actually you do not explicitly define NIR and SWIR in terms of wavelength range. This would be helpful. Or lose the terms entirely and just use the wavelengths.

We chose to refer to the SEVIRI band as visible, NIR and SWIR channel in order to easily compare with other satellite instruments such as MODIS. In the manuscript, the following sentences have been modified:

*The SEVIRI instrument, aboard the MSG satellite (Aminou et al., 1997), has channels centred at 0.64, in the visible, and at 0.81µm, in the NIR.*

*As in the Nakajima and King technique (1990), the sensitivity of the retrieval to the CER is brought by the Short-Wave Infra-Red (SWIR) channel of SEVIRI, centred at 1.64µm.*

Line 166: please explain 'two-way transmittance' – from where to where? Why is the two-way transmittance important?

The following information has been added to the manuscript:

*(i.e. from the top of the atmosphere to the cloud top and from the cloud top to the top of the atmosphere)*

*For sake of simplicity, the two-way transmittances will be referred to as transmittances.*

Line 228-233: Not really enough detail on 'weighting'. Someone would struggle to replicate what you have done from this info alone.

The explanation about the weighting of the fit has been rephrased:

*In order to obtain the most suitable aerosol optical parameters for the retrieval, it is important to accurately fit the PCASP measurements where the aerosols contribute the most to the SEVIRI signal. Each bin of the PCASP has been assigned a weight for the fit of the bimodal distribution. The weights have been calculated in a similar way to Haywood et al. (2003), which means that they are proportional to the contribution of each bin to the total aerosol extinction in the 0.6 µm band. The bins corresponding to the 0.15 to 0.25 µm radius range contribute to about 77% of the extinction. Consequently, these bins have been assigned appropriate larger weights during the fitting process of the size distribution.*

Line 255: For the uninitiated it might be useful to say where SAFARI was.

It is now mentioned that the SAFARI and the DABEX measurements were performed over the SEAO.

Line 309: 'typically observed in this region' – as shown by who exactly?

We have added the reference to Szczodrak et al. (2001).

*The cloud properties retrieved are within the range of values typically observed for marine stratocumulus (Szczodrak et al., 2001) with more than 90 % of the COT lower than 25 and 99 % of the CER between 4 and 20 μm.*

Szczodrak, M., Austin, P. H. and Krummel, P. B.: Variability of Optical Depth and Effective Radius in Marine Stratocumulus Clouds, J. Atmos. Sci., 58(19), 2912–2926, doi:10.1175/1520-0469(2001)058<2912:VOODAE>2.0.CO;2, 2001.

Line 322: Can you provide a reference for this statement please.

This sentence has been replaced by:
*These errors are likely upper estimates because forecast errors in specific humidity are unlikely to reach these values owing to the extensive assimilation of satellite data and sonde profiles by the data assimilation process used in the Met Office forecast model as previously mentioned.*

Line 473-474: This isn't immediately obvious to me. Can you clarify? Obviously you could use a different aerosol model in the LUT but this wouldn't be 'easy'.

In the case where the retrieved AAOT does not depend on the aerosol properties assumed for the retrieval, the AOT retrieved by SEVIRI could be converted from the aerosol model used for the retrieval to another aerosol model using their SSA. To emphasise the importance of the SSA assumption, this sentence has been replaced by the following text in the conclusion:
*This uncertainty is led by the sensitivity of the retrieval to the SSA. Because the method relies on the impact of the aerosol absorption on the light reflected by the clouds, the perturbation of the SSA has primarily an impact on the scattering contribution of the AOT. Therefore, a better accuracy is obtained on the retrieved AAOT, with biases generally lower than 17% before 15:00 UTC. After that time, an increase of the uncertainty on both the AOT and the AAOT has been observed, and users are advised to be careful when using the late afternoon aerosol product. For any satellite retrievals based on the colour-ratio technique, aerosol properties, including the SSA, have to be assumed and the same order of magnitude can be expected on the sensitivity of their AOT. This analysis highlights the importance of a suitable constrain on the SSA.*

Figure 5(b): You have lost the latitude labels

Figure 7-11: y-axes labels. Suggest adding 1-1 lines.

Figure 12: Add time basis (e.g. UTC).

These corrections have been applied.

[revised manuscript text omitted]

Figure 8 and 9 show the impact of a variation of +/-0.01 µm on the fine mode radius and +/-0.1 on the fine mode standard deviation. For each aerosol and cloud property, a linear relationship is observed between the retrieval using the standard CLARIFY-2017 aerosol model and the modified one. The aerosol size distribution has little influence on the retrieved cloud properties. On average, the modification of the fine mode standard deviation leads to a difference of 2.2% on the COT and 1.0% the CER. The effect associated with a change in the fine mode radius is even lower than 1%. As expected, the above-cloud AOT is more sensitive to the aerosol size distribution used for the inversion and differences up to 11.8% have been observed when the fine mode standard deviation is decreased by 0.1. However, the retrieval of the AOT is based on the detection of the aerosol absorption of the light reflected by the clouds. Therefore, the impact of an error on the aerosol size distribution on the AAOT retrieval is reduced to 5.4% for the standard deviation and 1.4% for the fine mode radius.

To assess the impact of the assumed aerosol refractive index on the retrieved aerosol and cloud properties of interest, variations of +/-0.02 and +/-0.008 have been applied to the real and imaginary parts of the refractive index, respectively. Figure 10 and 11 compare the retrieved aerosol and cloud properties from SEVIRI radiance data for the CLARIFY-2017 aerosol model with those retrieved when the aerosol refractive index parameters are perturbed. The influence of refractive index is similar to the one of the modified aerosol size distribution in that differences of <1% are observed in both COT and CER and a larger impact is found on the AOT with differences up to 39% where the imaginary refractive index is decreased by 0.008. The magnitude of the impact on the AOT is correlated to the difference of SSA between the CLARIFY-2017 and the perturbed aerosol model. Therefore, the retrieval of the AAOT is also less sensitive to the assumption on the aerosol refractive index, with an impact lower than 6.5%.

In order to evaluate the uncertainty $u_{aer}$ of the retrieved aerosol and cloud properties due to the aerosol model assumptions, we combined the uncertainty $u_i$ from the above sensitivity studies using:
The uncertainty has been estimated at 31.2% on the AOT, 2.3% on the COT and 1.2 % on the CER. Owing to the sensitivity of the retrieval to the aerosol absorption above clouds, a 6.1% uncertainty has been obtained on the AAOT, which is, together with the cloud albedo, the main parameter for the estimation of the DRE of absorbing aerosols above clouds.

| Page 28: [2] Deleted | Authors | 09/05/2019 12:56:00 |

| Page 28: [2] Deleted | Authors | 09/05/2019 12:56:00 |

| Page 28: [2] Deleted | Authors | 09/05/2019 12:56:00 |

| Page 28: [2] Deleted | Authors | 09/05/2019 12:56:00 |

| Page 28: [2] Deleted | Authors | 09/05/2019 12:56:00 |

**10:** Similar to Figure 8 for the impact

**10:** Similar to Figure 8 for the impact

**10:** Similar to Figure 8 for the impact

**10:** Similar to Figure 8 for the impact

[Figure]

**Figure 12:** Time series